# Mother-child dyadic interactions shape the developing social brain and Theory of Mind in young children

Lei Li[1,2], Jinming Xiao[1,2], Weixing Zhao[1,2], Qingyu Zheng[3], Xinyue Huang[1,2], Xiaolong Shan[1,2], Yating Ming[1,2], Peng Wang[1,2], Zhen Wu[4], Huafu Chen[1,2], Vinod Menon[5], Xujun Duan[1,2]*

[1]The Clinical Hospital of Chengdu Brain Science Institute, School of Life Science and Technology, University of Electronic Science and Technology of China, Chengdu, China; [2]MOE Key Lab for Neuro information, High-Field Magnetic Resonance Brain Imaging Key Laboratory of Sichuan Province, University of Electronic Science and Technology of China, Chengdu, China; [3]School of Healthcare Technology, Chengdu Neusoft University, Chengdu, China; [4]Department of Psychological and Cognitive Sciences, Tsinghua University, Beijing, China; [5]Departments of Psychiatry & Behavioral Sciences, Neurology & Neurological Sciences, Wu Tsai Neuroscience Institute, Stanford University School of Medicine, Stanford, United States

*For correspondence: duanxujun@uestc.edu.cn

## eLife Assessment

This **important** study reports **solid** evidence for the significant role of mother-child neural synchronization and relationship quality in the development of Theory of Mind (ToM) and social cognition. The findings effectively bridge brain development with children's behavior and parenting practices, and will be of interest to researchers studying brain development and social cognition, as well as the general public.

**Abstract** Social cognition develops through a complex interplay between neural maturation and environmental factors, yet the neurobehavioral mechanisms underlying this process remain unclear. Using a naturalistic fMRI paradigm, we investigated the effects of age and parental caregiving on social brain development and Theory of Mind (ToM) in 34 mother-child dyads. The functional maturity of social brain networks was positively associated with age, while mother-child neural synchronization during movie viewing was related to dyadic relationship quality. Crucially, parenting and child factors interactively shaped social cognition outcomes, mediated by ToM abilities. Our findings demonstrate the dynamic interplay of neurocognitive development and interpersonal synchrony in early childhood social cognition, and provide novel evidence for neurodevelopmental plasticity and reciprocal determinism. This integrative approach, bridging brain, behavior, and parenting environment, advances our understanding of the complex mechanisms shaping social cognition. The insights gained can inform personalized interventions promoting social competence, emphasizing the critical importance of nurturing parental relationships in facilitating healthy social development.

## Introduction

Social cognition, defined as the ability to interpret and predict others' behavior based on their beliefs and intentions and to interact in complex social environments and relationships is a crucial aspect of human development (*Adolphs, 2009*; *Frith and Frith, 2012*). Children's social cognitive abilities have far-reaching implications for their social competence, peer acceptance, and success in school (*Devine and Hughes, 2014*; *Milligan et al., 2017*). A key component of social cognition is the ability to understand and reason about others' mental states, known as Theory of Mind (ToM; *Adolphs, 2003*). ToM enables children to predict and interpret others' actions based on an understanding of their unobservable mental states, such as beliefs, desires, and emotions (*Wellman, 2014*). The development of ToM follows a predictable sequence throughout childhood, with major milestones occurring between the ages of 3 and 5 years (*Wellman, 2014*). During this period, children show a marked improvement in their ability to appreciate that others can hold beliefs and knowledge states that differ from their own (*Wimmer and Perner, 1983*). This understanding is typically assessed using false belief tasks, which require children to reason about what another person might mistakenly think or believe (*Fletcher et al., 1995*). Success on these tasks indicates a more adult-like understanding of the mind and is considered a hallmark of ToM development (*Kuhn and Siegler, 2006*).

Advances in neuroimaging techniques have allowed researchers to investigate the neural basis of social cognition and its development (*Vogeley et al., 2001*). Studies in adults have consistently identified a network of brain regions involved in ToM reasoning, including the medial prefrontal cortex, temporoparietal junction, precuneus, temporal lobes, and inferior frontal gyri (*Schurz et al., 2014*). Additionally, a distinct network, known as the social pain matrix (SPM), has been implicated in processing others' physical sensations and pain. This network comprises regions such as the bilateral insula, medial frontal gyrus, secondary somatosensory cortex, and anterior middle cingulate cortex (*Zaki et al., 2016*). ToM and SPM networks are often jointly recruited to reason about different kinds of internal states: the internal states of others' minds, including beliefs and desires (*Gallagher et al., 2000*), and emotions (*Mehnert et al., 2017*), and the internal states of others' bodies, including pain (*Bruneau et al., 2012*). Developmental studies have begun to chart the trajectory of these social brain networks in children, revealing that they become increasingly specialized and differentiated throughout childhood (*Jacoby et al., 2016*; *Richardson et al., 2018*; *Richardson, 2019*). For example, a recent study by Richardson et al. found that the ToM and SPM networks are functionally distinct by age 3 and show increasing within-network correlations and anti-correlations between networks from ages 3–12 (*Richardson et al., 2018*). These findings suggest that the development of specialized brain regions for reasoning about others' mental states and physical sensations is a gradual process that continues throughout childhood.

Crucially, the development of social cognition is thought to be influenced not only by the maturation of neural circuits with age, but, just as importantly, by social-environmental factors such as parental caregiving and parent-child relationship quality. Family is considered the primary socialization context for children, and parenting plays a crucial role in shaping a child's experiences and social development (*Belsky and de Haan, 2011*; *Hari et al., 2015*). Theories of bio-behavioral synchrony (*Feldman, 2012*) and social learning (*McLeod, 2011*) emphasize the importance of parent-child interactions and shared experiences in the development of social skills. Bio-behavioral synchrony refers to the matching of behavioral, physiological, and neural responses between mother and child during social interactions (*Feldman, 2012*). This synchronization is thought to facilitate emotional sharing and social understanding and has been linked to positive developmental outcomes (*Wheatley et al., 2012*; *Koole and Tschacher, 2016*). Bandura's model of reciprocal determinism, which highlights the bidirectional interactions between individuals and their environment, provides a framework for understanding how children acquire social skills through observation, imitation, and reinforcement (*Bandura, 1978*). Repeated experiences of social synchrony during parent-child interactions are believed to shape the developing brain and have long-lasting effects on children's social competencies (*Feldman, 2007*; *Feldman, 2010*). This dynamic interplay shapes their ability to understand and predict the behavior of others, which is crucial for the development of ToM and other social competencies.

One promising avenue for studying the neural underpinnings of parent-child interactions is the investigation of neural synchronization between parent and child during shared experiences. Naturalistic paradigms, such as movie viewing, offer several advantages over traditional fMRI tasks in studying

social brain development (*Finn, 2021*; *Vanderwal et al., 2019*). They require minimal instruction, are easily replicable across research sites and social contexts, and can be tailored in length to engage young children and reduce the confounding effects of head motion during fMRI scanning (*Hasson et al., 2010*; *Cantlon and Li, 2013*; *Vanderwal et al., 2015*). Moreover, they allow for the investigation of social processing in a more ecologically valid context, as movies often depict complex social interactions and relationships that closely resemble real-life experiences (*Nguyen et al., 2019*; *Li et al., 2022*). Previous studies have successfully used movie-viewing paradigms to investigate ToM and social pain processing in children (*Richardson et al., 2018*; *Richardson, 2019*), demonstrating the feasibility and reliability of this approach in developmental populations.

Previous studies have suggested that in parent-child dyads neural similarity varied depending on family connectedness, such that only dyads reporting high family connectedness showed similar neural profiles (*Lee et al., 2018*), and that atypical child-parent neural synchrony in medial prefrontal-hippocampus circuitry is associated with psychopathology in children (*Su et al., 2022*). However, the neural basis of parent-child behavioral synchrony and its relation to social cognitive outcomes remains largely unexplored. Moreover, few studies have taken an integrative approach to examine the interplay between intrinsic and environmental factors in shaping social brain development.

Here, we aim to address these gaps in the literature by investigating the neurobehavioral implications of intrinsic development and parental caregiving for children's social cognition. Understanding how these networks differentiate with age is essential not only for mapping typical brain development, but also for contextualizing the role of environmental influences. By establishing normative patterns of neural maturity and differentiation, we can better interpret how relational experiences—such as caregiver-child synchrony and parenting quality—modulate these trajectories. Thus, our first goal provides a developmental anchor that grounds our investigation of interpersonal and environmental contributions to social brain function. We examine developmental changes in the functional neural maturity of social brain networks and its relation to children's physiological age. We hypothesized that the neural maturity of the ToM and SPM networks, as measured by the similarity of children's brain responses to those of adults, would be positively associated with age. Our second goal was to investigate neural synchronization between child-mother dyads during a naturalistic movie-viewing paradigm and its association with parent-child relationship quality. We predicted that child-mother dyads would exhibit higher levels of inter-subject neural synchronization compared to child-stranger dyads and that the degree of synchronization would be positively related to the quality of the parent-child relationship. Our third goal was to explore the interaction between neurobehavioral factors of parenting and personal growth in predicting children's social cognition outcomes. We hypothesized that parenting factors, such as parent-child relationship quality and parental rearing behavior, would interact with personal growth factors, such as age and neural maturity, to predict children's ToM performance and social cognition deficits, with ToM serving as a mediator between these factors and social outcomes.

To test these hypotheses, we initially collected fMRI data from 50 child-mother dyads while they viewed an animated movie that depicted characters' mental states and physical sensations. 16 were excluded based on predefined data quality criteria, including excessive head motion and incomplete scans. The remaining 34 dyads constituted the final sample used for all analyses. We used reverse correlation analysis to identify movie events that drove activity in ToM and SPM brain networks, inter-region correlation analysis to assess the functional neural maturity of these networks, and inter-subject correlation analysis to measure neural synchronization between child-mother and child-stranger dyads. Structural equation modeling (SEM) was employed to examine the relationships among neurobehavioral factors of parenting and personal growth, ToM performance, and social cognition outcomes.

Our findings revealed that the functional neural maturity of social brain networks was positively associated with children's physiological age, providing evidence for the continued development and refinement of these networks throughout childhood. Moreover, we found that child-mother dyads exhibited higher levels of neural synchronization compared to child-stranger dyads and that the degree of synchronization was negatively associated with parent-child conflict. These results suggest that shared neural responses during naturalistic viewing may reflect the quality of the parent-child relationship and highlight the importance of considering dyadic factors in studying social brain development. Finally, SEM revealed that parenting and personal growth factors interacted to predict children's social cognition outcomes, with ToM performance serving as a mediator. Specifically, we found

**Table 1.** Relationship between individual behavioral characteristics and Theory of Mind (ToM) behavior.

| | Coefficient β | SE | P_Regress | R_Pearson | P_Pearson |
|---|---|---|---|---|---|
| Receptive Vocabulary (PPVT) | 0.203 | 0.030 | 0.206 | **0.434** | **0.010** |
| Social Responsiveness (SRS Total) | −0.295 | 0.040 | 0.074 | **−0.598** | **<0.001** |
| Parent-Child Conflict (CPRS) | −0.191 | 0.103 | 0.189 | **−0.377** | **0.028** |
| Parent-Child Closeness (CPRS) | 0.164 | 0.061 | 0.264 | 0.265 | 0.130 |
| Parental Rejection (EMBU) | 0.296 | 0.151 | 0.075 | **0.388** | **0.023** |
| Parental Emotional Warmth (EMBU) | 0.258 | 0.100 | 0.184 | 0.056 | 0.753 |
| Parental Control Attempts (EMBU) | 0.170 | 0.104 | 0.210 | 0.148 | 0.405 |
| Child Age | 0.348 | 0.577 | 0.073 | **0.434** | **0.010** |

Note: $\beta$ is the standard regression coefficient from multiple regression analysis. P_Regress is the P value from multiple regression analysis. R_Pearson is the Pearson correlation coefficient from correlation between the individual behavioral characteristics and ToM behavior. P_Pearson is the P value from correlation between the individual behavioral characteristics and ToM behavior. SRS = Social Responsiveness Scale; CPRS = Child–Parent Relationship Scale; EMBU = Egna Minnen av Barndoms Uppfostran; PPVT = Peabody Picture Vocabulary Test. Bold font indicates the significance level at $P<0.05$.

that parenting factors had an indirect effect on social cognition deficits through their influence on children's personal growth and ToM abilities. These findings provide novel insights into the neuro-behavioral mechanisms underlying social cognition development and highlight the importance of considering both intrinsic and environmental factors in shaping social brain function.

## Results

### The association between ToM behavior and individual behavioral characteristics

A total of 100 participants (50 mother-child dyads, with children between the ages of 3–8 years) participated in the study. Sixteen dyads were excluded due to high levels of head motion during fMRI scanning. Data from the remaining 34 mother-child dyads were used in further neuroimaging analysis.

All children completed a behavioral battery after the fMRI scan, which included a custom-made explicit ToM task (see Methods). Multiple regression and correlation analyses were used to evaluate the relationship between performance on the ToM task (i.e. proportion correct, referred to as ToM behavior) and general demographic and behavioral measures. Comparison between included and excluded dyads revealed no significant differences in child age (t=0.78, p=0.44), ToM scores (t=0.92, p=0.36), or gender distribution ($\chi^2$=0.48, p=0.49), indicating that data exclusion did not bias the sample in a systematic way. Significant associations between ToM behavior and age (r=0.434, p=0.027, false discovery rate [FDR]-corrected), conflict on Child–Parent Relationship Scale (CPRS) (r=−0.377, p=0.048, FDR-corrected), rejection on the EMBU (r=0.388, p=0.048, FDR-corrected) (Egna Minnen Betriiffande Uppfostran, a self-report questionnaire to obtain ratings from parents about their own rearing behavior with their children), standard score on Peabody Picture Vocabulary Test, Fourth Edition (PPVT-4) (r=0.434, p=0.027, FDR-corrected) and total scores on social responsiveness scale (SRS) (r=−0.598, Pp<0.008, FDR-corrected) were detected. No significant relationships were observed between ToM behavior and other behavioral characteristics (*Table 1*).

### Reverse correlation analysis

We used reverse correlation analyses to identify events lasting more than 4 s in the continuous naturalistic stimulus that evoked reliable positive hemodynamic responses in the same brain region across subjects (*Hasson et al., 2004*). Reverse correlation analyses were performed on the average response time courses in each network to identify events that drive activity in ToM and SPM brain regions (*Figure 1A*). The regions of interest (ROIs) for the ToM and SPM networks were based on a previously published study (*Table 2*; *Richardson et al., 2018*).

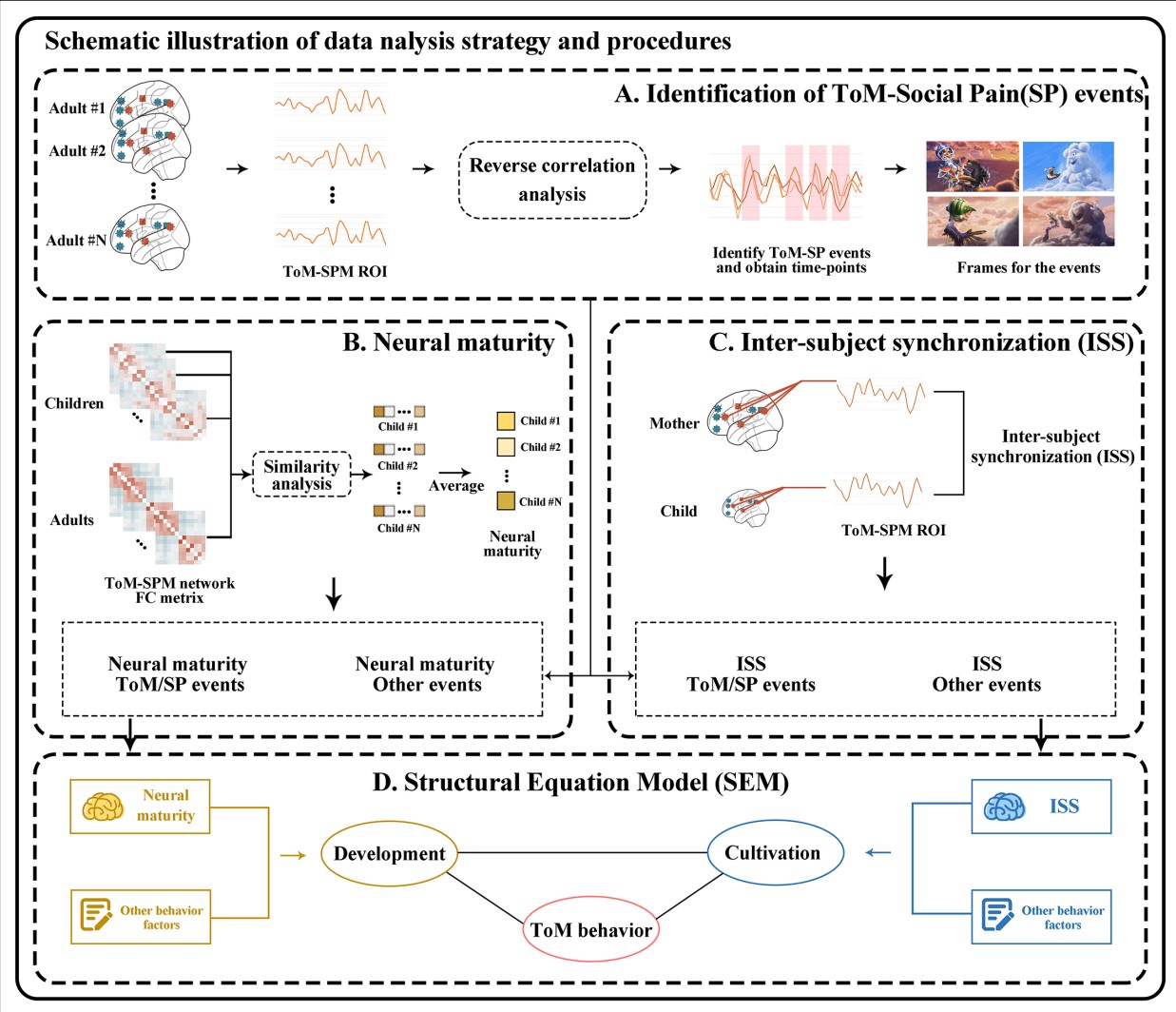

**Figure 1.** Overview of analysis pipeline. (**A**) Reverse correlation analysis was conducted on the average response network timecourses to identify ToM and Social Pain events driving activity in Theory of Mind (ToM) and Social Pain Matrix (SPM) related brain regions. (**B**) First, inter-region correlations were computed across all ToM and SPM brain regions of interest for each participant. Neural maturity of a child was then assessed by averaging the similarity between the child's correlation matrices and those of each adult. (**C**) Inter-subject synchronization (ISS) was determined by calculating the correlation of neural response time series between child-mother and child-stranger dyads. (**D**) A structural equation model was employed to explore the relationships among neurobehavioral factors of parenting and personal growth, ToM performance, and social cognition outcomes.

In adults, reverse correlation analysis identified seven ToM events (60 s total, length 8.3±6.18 s) and eight social pain events (52 s total, length 6.5±2.60 s). All seven peak 'mind' events depicted the characters' beliefs, desires, and/or emotions (e.g. Peck finds other clouds and cranes are happy, Baby cries and then is made to laugh) (*Figure 2A*). All eight peak 'social pain' events depicted characters experiencing social pain (e.g. Peck being bitten by a hedgehog) or events such as thunder and lightning (*Figure 2B*). The five events that had the highest response magnitude in each network in adults are shown in *Figure 2C*; see *Appendix 2—table 1* for full descriptions of these events, including timing and duration information. The timepoints that exceeded baseline for ToM and SPM networks were almost entirely non-overlapping, with the exception of a single timepoint (2 s). Timepoints corresponding to ToM and Social Pain events were defined as ToM/SP events (132 TRs), while other timepoints were defined as other events (168 TRs) for further analysis.

Additionally, reverse correlation analysis conducted on the children alone identified 6 of the 8 social pain events and 4 of the 7 ToM events discovered in the adult sample (*Figure 2A–B*). This

**Table 2.** Definition of ToM and SPM regions of interest.

Regions identified, center coordinate [x y z] for each region of interest in the Theory of Mind (ToM) and Social Pain Matrix (SPM) networks. RTPJ and LTPJ, right and left temporoparietal junction; PC, precuneus; DMPFC, MMPFC and VMPFC, dorsal, middle, and ventral components of medial prefrontal cortex; RS2 and LS2, right and left secondary sensory; Rinsula and Linsula, right and left insula; RMFG and LMFG, right and left middle frontal gyrus; AMCC, anterior middle cingulate cortex.

| Network | ROI | Center coordinate |
|---|---|---|
| ToM | RTPJ | (48 -60 30) |
| | LTPJ | [-48–62 30] |
| | PC | [0–54 34] |
| | DMPFC | [–6 54 36] |
| | MMPFC | [–4 58 16] |
| | VMPFC | [–4 56–16] |
| | | |
| Pain | RS2 | (60 -28 38) |
| | LS2 | [-62–32 34] |
| | Rinsula | (42 6 -6) |
| | Linsula | [-42–2 –4] |
| | RMFG | (50 42 12) |
| | LMFG | [–46 36 14] |
| | AMCC | [0 2 42] |

indicates some overlap in the neural processing of social pain and ToM events between children and adults, highlighting the potential developmental continuity in these neural networks.

## Neural maturity reflects the development of social brain

Children were divided equally into three groups according to their age (*Table 3*): Pre-junior group (3.3–4.8 years), Junior group (5.0–5.8 years), and Senior group (6.0–8.0 years). Average correlation matrices were computed for different age groups to reveal the extent to which a group of brain regions operate as a network with synchronized responses. In adults, each network exhibited strong positive correlations within-network and negative correlations across networks within-ToM correlation M(s.e.)=0.51 (0.04); within-SPM correlation M(s.e.)=0.39 (0.05); across-network M(s.e.) = −0.11 (0.01). In children, each network also exhibited positive correlations within-network and negative correlations across networks within-ToM correlation M(s.e.)=0.31 (0.04); within-SPM correlation M(s.e.)=0.29 (0.04); across-network M(s.e.) = −0.09 (0.02). The pattern of network correlations exhibited substantial strengthening between the ages of 3 and 8 years (*Figure 3A*). Among children, average correlations within-ToM (*r*=0.40, *p*=0.02, FDR-corrected) and within-SPM network (*r*=0.32, *p*=0.05, FDR-corrected) increased significantly with age (*Figure 3B*). Nevertheless, the two networks were already functionally distinct in the youngest group of children we tested. In the Pre-junior group only (3–4 years old children, n=12), both ToM and SPM networks had positive within-network correlations within-ToM correlation M (s.e.)=0.29 (0.06); within-SPM correlation M(s.e.)=0.23 (0.05), across-network M(s.e.) = −0.05 (0.02).

Next, we tested whether the functional neural maturity (i.e. similarity of correlation matrices to adults) of a child was related to their physiological age or parent-child relationship quality. The neural maturity of each child was quantified by averaging the similarity between their correlation matrices and those of each adult within the ToM/SPM networks (*Figure 1B*). Our findings revealed a significant correlation between neural maturity and age (*r*=0.42, *p*=0.01, FDR-corrected), while no significant correlation was observed with parent-child relationship quality (*r*=–0.13, *p*=0.47) (*Figure 3C–D*). Furthermore, we observed a significant correlation between neural maturity and age during ToM/SP events (*r*=0.46, *p*=0.01, FDR-corrected), while no such correlation was found during other events

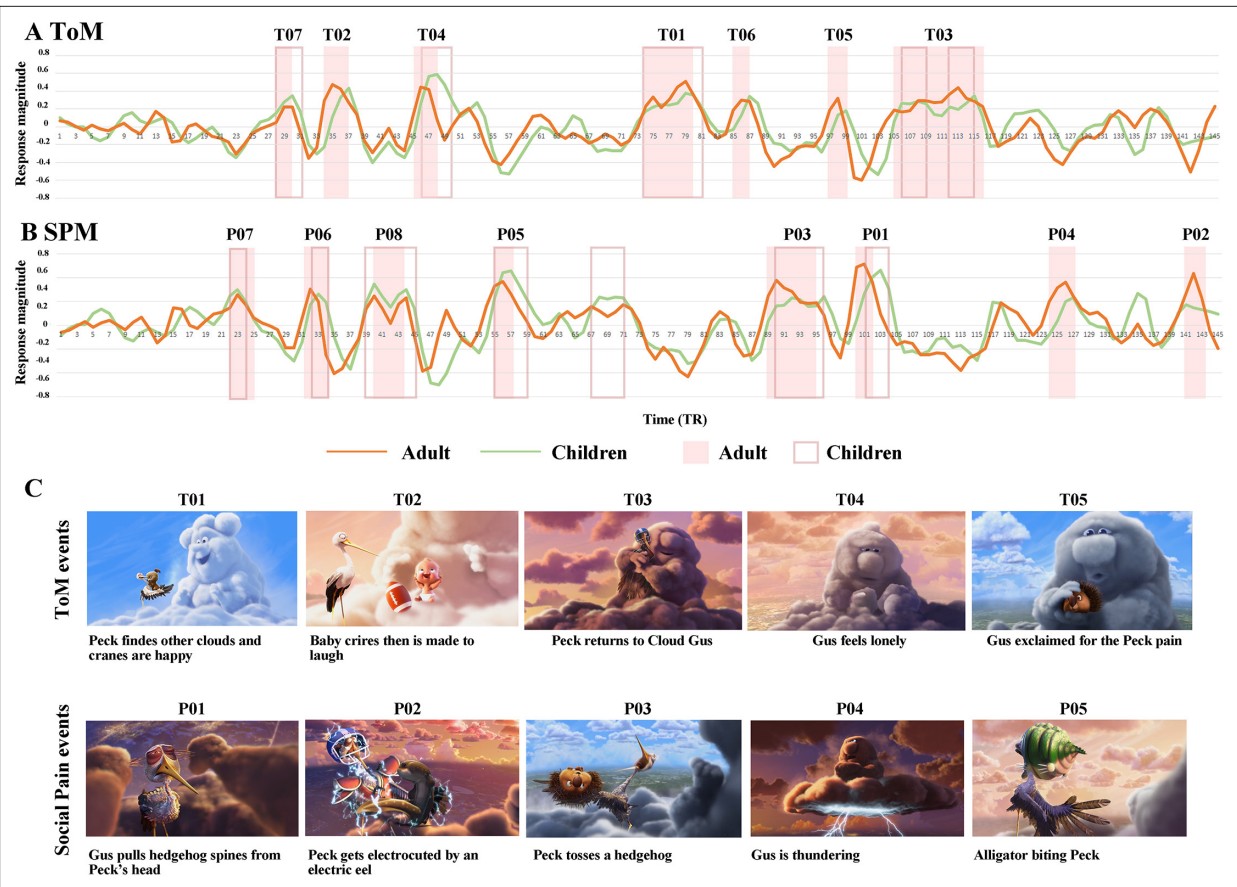

**Figure 2.** Reverse correlation analysis. The average timecourses of child (green) and adult (red) groups for the (**A**) Theory of Mind (ToM) and (**B**) Social Pain Matrix (SPM) networks during movie viewing are presented. Each time point along the x-axis corresponds to a single repetition time (TR) (2 s). Shaded blocks represent time points identified as ToM and Social Pain events in a reverse correlation analysis conducted on adults, while dark borders indicate time points identified as ToM and Social Pain events in children. Event labels (e.g. **T01**, **P01**) denote the ranking of average response magnitude in adults. (**C**) Example frames and descriptions for the five events with the highest response magnitude in adults are provided.

($r=–0.04$, $p=0.84$) (*Figure 3E*). A paired $t$-test revealed that the neural maturity of children during ToM/SP events was lower than that during other events (T=–6.29, $p<0.01$). A direct comparison revealed a significant difference in neural maturity between ToM/SP events and age for ToM/SP versus other events (F=3.48, $p=0.046$, *Figure 3F*).

## Inter-subject neural synchronization and child-parent relationship quality

First, we examined whether the inter-subject synchronization (ISS) between child-mother dyads within the ToM/SPM network was associated with parent-child relationship quality. We averaged fMRI timeseries across all ToM and SPM ROIs to obtain a network-level timeseries. ISS was then defined as the temporal correlation between the mean fMRI time series of the child-mother dyads or child-stranger dyads, and a Fisher's r-to-z transform was applied to convert the correlation coefficient (r) value to a normally distributed variable z (*Figure 1C*). Compared to child-stranger dyads, ISS was higher in child-mother dyads (Two-sample $t$ tests, T=2.48, $p=0.016$) (*Figure 4A*). Additionally, partial correlation analysis with the age of children as a covariate was performed to explore the relationship between ISS and child-parent relationship quality. A significant negative correlation between conflict scores on the CPRS and ISS was found within child-mother dyads ($r=–0.41$, $p=0.02$, FDR-corrected), while such correlation was not found within child-stranger dyads ($r=0.20$, $p=0.24$) (*Figure 4B*). Moreover, there was no significant correlation between age and either ISS within child-mother dyads or within child-stranger dyads (*Figure 4C*).

**Table 3.** Demographic information and behavioral data by age group.
Note: PPVT, Peabody Picture Vocabulary Test, Fourth Edition; CPRS = Child-Parent Relationship Scale; EMBU = Egna Minnen av Barndoms Uppfostran (parenting style); SRS = Social Responsiveness Scale.

| Group | N | Age (Range) | Gender (Female/Male) | Handedness (Right/Left/Mix) | TOM behavior score | PPVT-4 (Standard score) | Parents-children relationship (CPRS) | | Parents' rearing behavior (EMBU) | | | Social Responsiveness (SRS) Total |
|---|---|---|---|---|---|---|---|---|---|---|---|---|
| | | | | | | | Conflict | Closeness | Rejection | Emotional warmth | Control attempts | |
| Pre-junior | 12 | 4.1±0.4 (3.3~4.8) | 6/6 | 11/0/1 | 7.4±4.7 | 139.2±18.8 | 43.3±5.8 | 22.5±9.0 | 16.4±4.46 | 59.4±7.0 | 43.1±5.0 | 36.7±15.0 |
| Junior | 11 | 5.5±0.3 (5.0~5.8) | 5/6 | 10/0/1 | 10.6±2.3 | 152.3±10.8 | 44.4±5.0 | 26.5±8.0 | 18.8±3.0 | 56.2±8.8 | 41.0±4.8 | 32.0±13.7 |
| Senior | 11 | 6.7±0.7 (6.0~8.0) | 6/5 | 10/0/1 | 11.8±1.9 | 151.5±19.0 | 40.2±3.8 | 27.5±9.33 | 19.2±3.4 | 56.5±5.1 | 42.0±4.6 | 23.0±13.8 |
| Adult (Mother) | 34 | 36.7±4.0 (30.7~48.8) | 34/0 | 33/0/1 | NA | NA | NA | NA | NA | NA | NA | NA |

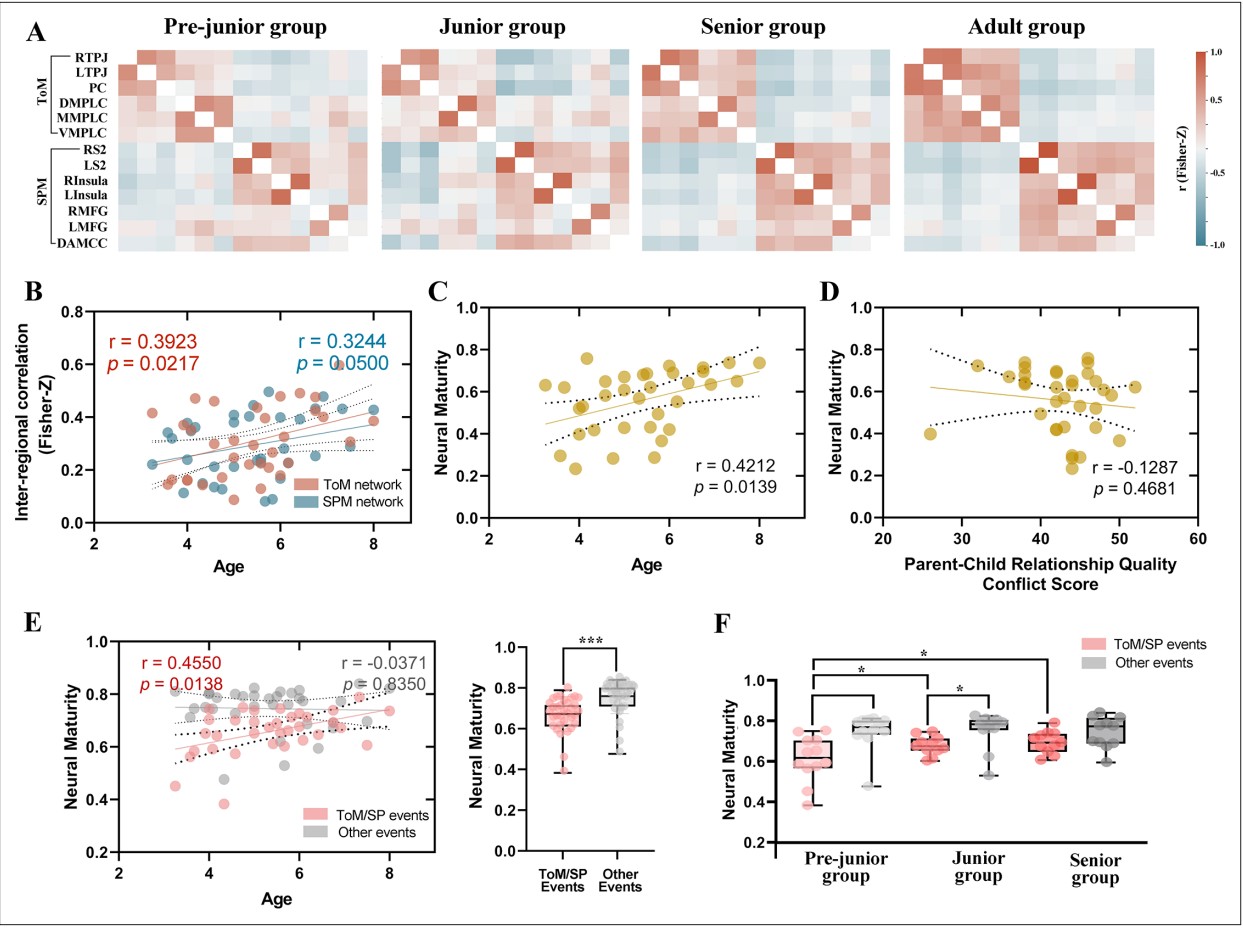

**Figure 3.** Inter-region correlation analysis and neural maturity. (**A**) Average z-scored correlation matrices were computed across all Theory of Mind (ToM) and Social Pain Matrix (SPM) regions of interest for each age group (Pre-junior: n=12; Junior: n=11; Senior: n=11; Adults: n=34). The nomenclature of brain regions is shown in *Table 2*. (**B**) Correlation between the average inter-regional correlation within ToM/SPM networks and age. (**C**) Correlation between neural maturity and age. (**D**) Correlation between neural maturity and conflict score of Child-Parent Relationship Scale (CPRS). (**E**) Correlation between neural maturity and age during ToM/SP and other events, as defined by reverse correlation analysis. Statistical comparisons of group differences (n = 34) were performed using two-sample *t*-tests. ***$p<0.001$. (**F**) Group differences in neural maturity during ToM/SP and other events for each age group (Pre-junior: n = 12; Junior: n = 11; Senior: n = 11). Statistical comparisons were conducted using two-sample *t*-tests. *$p<0.05$.

Subsequently, we investigated whether the association between relationship quality and ISS was specific to ToM/SP events. ANOVA revealed a main effect of parent-child relationship (child-mother dyads vs. child-stranger dyads), with stronger ISS in child-mother dyads ($F=8.84$, $p=0.004$). ANOVA also revealed a main effect of events (ToM/SP vs. other events), with stronger ISS during ToM/SP events ($F=10.05$, $p=0.002$). No significant interaction effect was observed between events and parent-child relationship ($F=0.10$, $p=0.75$). Partial correlation indicated that ISS between child-mother dyads was marginally significantly correlated with conflict scores during ToM/SP events ($r=-0.34$, $p=0.05$, FDR-corrected), while such a correlation was not present during other events ($r=0.03$, $p=0.86$) (*Figure 4D*). No such significant correlation was observed in child-stranger dyads (*Figure 4E*). In comparison to other events, ISS was found to be higher during ToM/SP events for both child-mother and child-stranger dyads (Pair *t*-tests, $p<0.05$, FDR-corrected).

## Structural equation modeling of parenting, personal traits, and ToM factors underlying children's social skills

We used SEM to examine the relation between neurobehavioral factors and parenting and personal traits, including conflict/closeness from the CPRS, warmth/rejection/control from the EMBU, sex, age, neural maturity, and ISS between child-mother dyads. Additionally, verbal intelligence, measured by the PPVT-4 standard score, was also used in the model estimation, recognizing the indispensable role

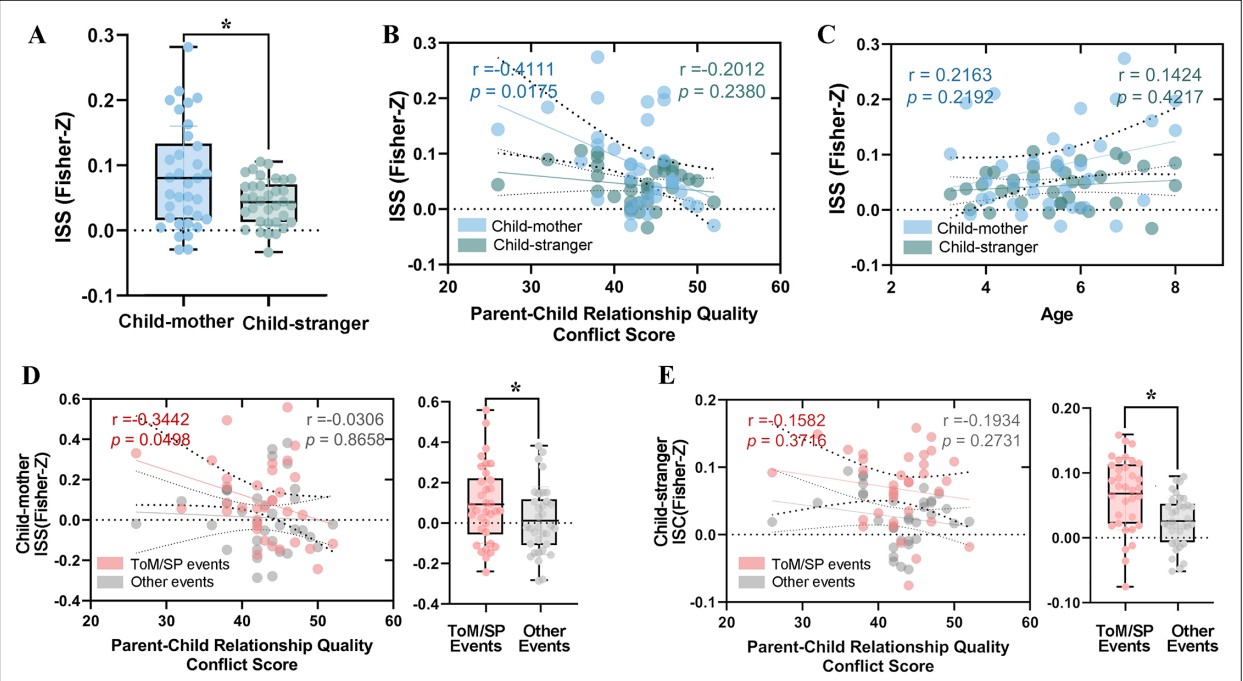

**Figure 4.** Inter-subject neural synchronization. (**A**) Group differences (n = 34) in inter-subject synchronization (ISS) during movie viewing. Statistical comparisons were performed using two-sample *t*-tests. (**B**) Partial correlation between conflict scores of Child-Parent Relationship Scale (CPRS) and ISS within child-mother and child-stranger dyads. (**C**) Correlation between age and ISS within child-mother and child-stranger dyads. (**D–E**) Partial correlation between conflict scores on the CPRS and ISS within child-mother and child-stranger dyads during ToM/SP events and other events. Group differences (n = 34) in ISS during ToM/SP and other events are displayed on the right. Statistical comparisons were conducted using two-sample *t*-tests. *\*p<0.05.*

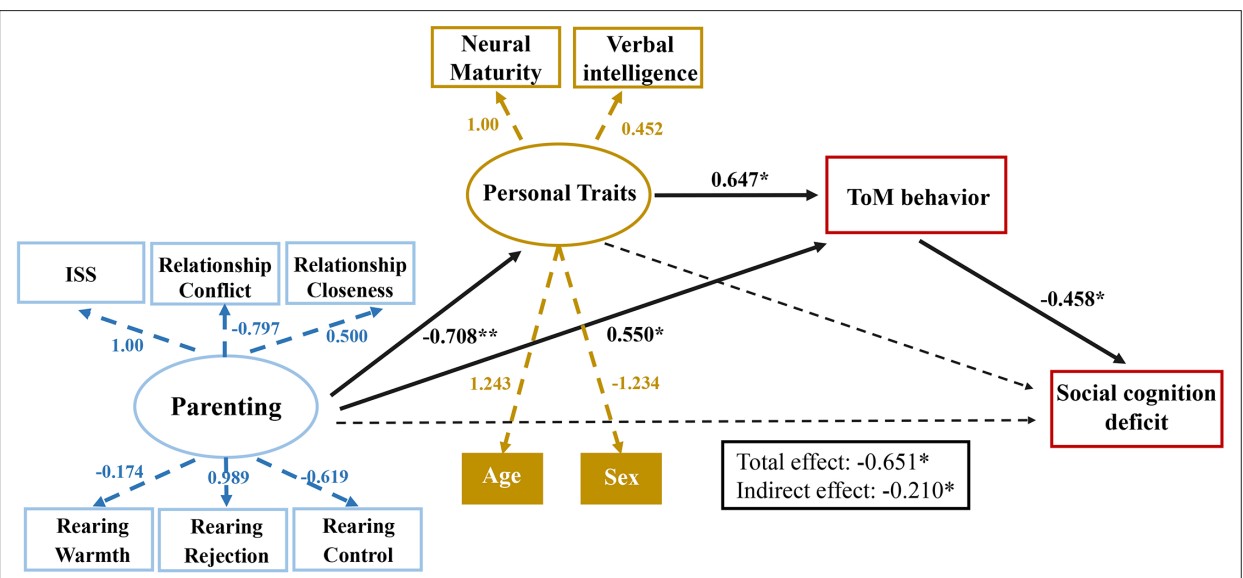

**Figure 5.** Structural equation model of latent personal traits, latent parental caregiving, ToM behavior, and social development outcomes. Parenting has a direct and indirect influence on ToM behavior which in turn influences social cognition. Latent factor underlying Personal Trait includes a child's neural maturity, while the latent factor underlying Parenting includes child-mother inter-subject neural synchronization (ISS). Regression coefficients are displayed for each path. Solid lines indicate significant paths (*\*p<0.05; \*\*p<0.01*), and dashed arrows indicate nonsignificant paths.

of language in the development of ToM (*Astington and Jenkins, 1999*). The SEM involved confirmatory factor analysis to consolidate the neurobehavioral factors into latent variable constructs and serial mediation analysis to explore the relationships between these latent constructs, ToM behavior, and social cognition deficits (measured by total SRS scores). The model showed a strong fit to the data, $\chi^2 = 14.16$, $\chi^2/df = 1.09$, CFI = 0.957, RMSEA = 0.052 (*Figure 5*).

As expected, child-parent relationship quality, rearing behavior, and child-parent ISS contributed significantly to the latent construct of *parenting* ($ps < 0.01$), and individual child's neural maturity, verbal intelligence, age, and sex contributed significantly to the latent construct of *personal trait* ($ps < 0.001$). *Parenting* had a significantly positive effect on ToM behavior and a negative effect on *personal trait*. *Personal trait* had a positive effect on ToM behavior, and ToM behavior had a negative effect on social cognition deficits. *Parenting* had a significant total effect on social cognition deficits ($\beta = -0.651$, $P < 0.05$). Additionally, a serial indirect mediating effect was found, where *parenting* and *personal trait* factors interacted to predict children's social cognition outcomes, with ToM performance serving as a mediator ($\beta = -0.210$, $p < 0.05$) (*Appendix 2—table 2*). Finally, additional control analyses revealed that both ToM and SPM networks are essential for predicting social cognitive outcomes (**Appendix 2**).

## Discussion

Our study aimed to elucidate the neurobehavioral mechanisms underlying the development of social cognition in children by examining the interaction between brain maturation with age and parental caregiving. Using a naturalistic movie-viewing paradigm and advanced neuroimaging analyses, we investigated the neural maturity of social brain networks, neural synchronization between child-mother dyads, and the interplay between neurobehavioral factors in predicting social cognition outcomes. We found that individual factors, such as age, were related to the functional neural maturity of social brain networks, social-environmental factors, such as parent-child relationship quality, were associated with parent-child neural synchronization during naturalistic viewing, and personal and social environmental factors interacted to predict children's social cognition outcomes, with ToM abilities serving as a mediator. Together, our findings support emerging theoretical frameworks emphasizing dyadic neural coupling and developmental plasticity, particularly the biobehavioral synchrony model and neuroconstructivist approaches to brain development. In addition, our findings align with Bandura's theoretical models of social learning, particularly the concept of reciprocal determinism, which posits that individual behavior is influenced by and influences personal factors and the social environment. Our study bridges the gap between theoretical perspectives and empirical research in developmental social neuroscience. This integration of theory and research is crucial for advancing our understanding of the complex processes that shape children's social cognitive abilities and for informing the development of effective interventions aimed at promoting social competence.

### Social brain neural network matures with age

We found that the functional neural maturity of social brain networks, specifically the ToM and SPM networks, was positively associated with children's physiological age. This relationship was specific to events depicting mental states and bodily sensations, suggesting that the maturation of these networks is tied to children's developing ability to reason about others' minds and experiences. The strengthening of within-network correlations and differentiation between the ToM and SPM networks with age further supports the notion that social brain development involves increasing specialization and refinement of neural circuits. While our primary analyses treated age as a continuous variable, we also performed exploratory group-based comparisons to probe for potential non-linear developmental shifts in social brain network organization. This approach revealed that the differentiation between ToM and SPM networks was already present in the youngest group (ages 3–4), suggesting that early neural specialization may begin prior to the age at which ToM behavior is reliably observed. These group-level observations provide complementary evidence to the continuous analyses and may inform future work examining sensitive periods or early markers of social brain development.

Our findings contribute to and extend prior research using fMRI paradigms to investigate ToM development in children. Previous work has shown that these networks become increasingly specialized and differentiated throughout childhood (*Richardson et al., 2018*; *Carter and Pelphrey, 2006*; *Cantlon et al., 2011*). The current study extends these findings by demonstrating that the

development of social brain networks is a gradual process that continues beyond the preschool years and is related to children's chronological age. This finding is consistent with behavioral research indicating that ToM and social abilities continue to develop and refine throughout middle childhood and adolescence (*Im-Bolter et al., 2016*). Importantly, we move beyond prior work by combining reverse correlation with naturalistic stimuli to isolate discrete, behaviorally meaningful events (e.g. mental state attribution, social rejection) and relate children's brain responses to adult patterns and social outcomes. This event-level analysis in a dyadic context offers greater ecological and interpretive precision than traditional block or condition-based designs. Our study provides novel evidence for the neural underpinnings of this protracted development, suggesting that the functional maturation of social brain networks may support the continued acquisition and refinement of social cognitive skills.

## Child-mother neural synchrony is linked to relationship quality

In parallel, our study builds on and extends a growing body of work on parent-child neural synchrony, much of which has relied on fNIRS or EEG hyperscanning to demonstrate interpersonal alignment during communication, shared attention, or cooperative tasks (*Deng et al., 2024*; *Miller et al., 2019*; *Nguyen et al., 2020*; *Reindl et al., 2018*). While these modalities offer fine temporal resolution, they are limited in spatial precision and typically focus on surface-level cortical regions such as the prefrontal cortex. By contrast, our naturalistic fMRI approach enables the examination of deep and distributed brain networks—specifically those supporting social cognition—within child-parent dyads during emotionally and cognitively rich scenarios. Intriguingly, we found that neural synchronization during movie viewing was higher in child-mother dyads compared to child-stranger dyads. Moreover, the degree of neural synchrony between children and mothers was specifically associated with lower parent-child conflict, but not children's age. This brain-to-brain coupling may facilitate the sharing of mental states and co-regulation of emotions between parents and children (*Nguyen et al., 2021*), thereby promoting social understanding. Our results corroborate previous behavioral studies linking mother-child synchrony to positive developmental outcomes (*Feldman, 2010*). They also extend prior neuroimaging work on interpersonal neural synchronization in relationships (*Endevelt-Shapira and Feldman, 2023*) by demonstrating that synchronous brain activity between parents and children is tied to the quality of their relationship. Notably, the association between neural synchrony and relationship quality was specific to events highlighting mental states and bodily sensations, underscoring the relevance of this neural coupling to children's developing social cognition. These findings support the theory of bio-behavioral synchrony (*Feldman, 2012*; *Bell, 2020*), which posits that synchronized behavioral, physiological, and neural responses between parent and child facilitate emotional sharing and social understanding, contributing to positive developmental outcomes.

## Parenting and biological factors impact social cognition via Theory of Mind

Our next goal was to determine how neurobehavioral factors of parenting and personal growth interact to predict children's social cognition outcomes. SEM revealed that parenting factors, such as the quality of the parent-child relationship and parental rearing behavior, significantly influenced children's personal neurocognitive growth and ToM abilities, which in turn impact social cognition. Specifically, parenting had an indirect effect on social cognition deficits through its influence on children's personal traits and ToM performance. These results highlight the importance of both intrinsic factors (age, neural maturity) and environmental factors (quality of parenting) in shaping the development of social cognition.

This finding accords with long-standing theories positing that children's understanding of mental states is shaped by both environmental input from caregivers and intrinsic neurocognitive maturation (*Hughes et al., 2005*; *Liu et al., 2008*). By representing these factors as latent variables, our analysis could capture their shared covariance while modeling their distinct pathways to ToM development. The results suggest that parenting and advantageous personal characteristics facilitate children's grasp of others' minds, which in turn supports broader social cognitive competencies.

The mediating role of ToM aligns with research on autism spectrum disorders, where impairments in mental state reasoning are thought to underlie many of the observed social cognition deficits (*Frith and Frith, 2003*; *Pellicano, 2010*). Interventions targeting ToM skills could thus potentially

ameliorate social difficulties in both neurotypical and clinical populations. Our findings highlight parenting behavior and parent-child relationship quality as promising targets for such family-based interventions.

## Insights into mechanisms of neuroconstructivist perspectives and Bandura's social learning theory

Our findings align with a neuroconstructivist perspective, which conceptualizes brain development as an emergent outcome of reciprocal interactions between biological constraints and context-specific environmental inputs. Rather than presuming fixed traits or linear maturation, this perspective highlights how neural circuits adaptively organize in response to experience, gradually supporting increasingly complex cognitive functions (*Westermann et al., 2007*). It offers a particularly powerful lens for understanding how early caregiving environments modulate the maturation of social brain networks.

Building on this framework, the present study reveals that moment-to-moment neural synchrony between parent and child, especially during emotionally salient or socially meaningful moments, is associated with enhanced Theory of Mind performance and reduced dyadic conflict. This suggests that beyond age-dependent neural maturation, dyadic neural coupling may serve as a relational signal, embedding real-time interpersonal dynamics into the child's developing neural architecture (*Karmiloff-Smith et al., 2018*). Our data demonstrate that children's brains are not merely passively maturing, but are also shaped by the relational texture of their lived experiences—particularly interactions characterized by emotional engagement and joint attention. Importantly, this adds a new dimension to neuroconstructivist theory: it is not simply whether the environment shapes development, but how the quality of interpersonal input dynamically calibrates neural specialization. Interpersonal variation leaves detectable signatures in the brain, and our use of neural synchrony as a dyadic metric illustrates one potential pathway through which caregiving relationships exert formative influence on the developing social brain.

The contribution of this work lies not in reiterating the interplay of nature and nurture, but in specifying the mechanistic role of interpersonal neural alignment as a real-time, context-sensitive developmental input. Neural synchrony between parent and child may function as a form of relationally grounded, temporally structured experience that tunes the child's social brain toward contextually relevant signals. Unlike generalized enrichment, this form of neural alignment is inherently personalized and contingent—features that may be especially potent in shaping social cognitive circuits during early childhood.

Although our study was not designed to directly examine learning mechanisms such as imitation or reinforcement, the findings can be viewed as broadly consistent with social learning theory. Bandura's theory posits that human behavior is shaped by observational learning and modeling from others in one's environment (*Bandura, 1978*; *Nabavi, 2012*; *Rumjaun and Narod, 2020*). According to Bandura, children acquire social cognitive skills by observing and interacting with their parents and other significant figures in their environment. This dynamic interplay shapes their ability to understand and predict the behavior of others, which is crucial for the development of ToM and other social competencies.

Aligned with this theory, we found that neural synchronization between children and their mothers was higher compared to child-stranger dyads. This increased neural coupling likely facilitates the sharing of mental states, emotions, and behaviors between parent and child, allowing children to learn and model social cognition skills through observing and mirroring their parents. The theory also highlights the importance of the social environment, particularly parents, in shaping behavior and cognitive development. Our results also revealed that higher parent-child neural synchrony was associated with lower mother-child conflict and better relationship quality. A supportive parenting environment with positive social interactions enables effective modeling and acquisition of social cognitive abilities in children. We also found that personal factors like children's neural maturity and age influenced the degree of neural synchronization with mothers. At the same time, parenting quality (relationship, rearing behavior, neural synchrony) impacted children's ToM development and social cognition abilities. Taken together, Bandura's social learning theory, with its emphasis on observational learning from social models, reciprocal influences between personal factors and the social environment, and the role of nurturing interactions, provides a robust framework for understanding the neurobehavioral mechanisms underlying the development of social cognition through parent-child neural synchronization.

## Limitations and future directions

While leveraging a naturalistic movie-viewing paradigm allowed us to study children's spontaneous neural responses during a semi-structured yet engaging task, dedicated experimental designs are still needed to make stronger inferences about the cognitive processes involved. Moreover, the ROIs used to define the ToM and SPM networks were based on meta-analyses and task studies primarily conducted with adults. While this approach promotes comparability with existing literature, it assumes that the spatial organization of these networks is stable across age groups. However, theories of interactive specialization suggest that the composition and boundaries of functional networks may undergo reorganization during development, with regions potentially entering or exiting networks based on experience and maturational processes. As a result, the current analysis may not fully capture age-specific functional architecture, particularly in younger children. Future studies using data-driven or age-appropriate parcellation methods could provide more precise characterizations of how social brain networks are constructed and differentiated throughout childhood. Additionally, our region-of-interest approach precluded examination of whole-brain networks; future work could explore developmental changes in broader functional circuits. The cross-sectional nature of our study is a further limitation, as it cannot definitively establish the causal directions of the observed relationships. Longitudinal designs tracking children's brain development and social cognitive abilities over time would help clarify whether early parenting impacts later neural maturation and behavioral outcomes, or vice versa. Another limitation of the current design is the lack of a resting-state baseline for inter-subject synchronization. While our focus was on synchronization during naturalistic social processing, we cannot determine whether individual differences in ISS reflect purely task-induced coupling or are partially shaped by trait-level synchrony present at rest. Including both resting and task conditions in future work would allow for stronger inferences about stimulus-specific versus baseline-driven synchronization, especially in relation to interpersonal factors such as relationship quality or social responsiveness. In addition, the modest sample size (N=34 dyads) presents limitations for the application of structural equation modeling (SEM), which typically requires larger samples for stable estimation and generalizable inferences. While the model fit was acceptable, the results should be interpreted as exploratory and hypothesis-generating, rather than confirmatory. Future studies with larger, independent samples will be important for validating the structure and directionality of the proposed relationships. Our sample was also restricted to mother-child dyads, leaving open questions about potential differences in father-child relationships and gender effects on parenting neurobiology (*Swain et al., 2014*). Larger and more diverse samples would enhance the generalizability of the findings.

Several future directions emerge from this research. First, combining naturalistic neuroimaging with structured cognitive tasks could elucidate the specific mental processes underlying children's neural responses during movie viewing. Examining how these processes relate to real-world social behavior would further bridge neurocognitive function and ecological validity. Longitudinal studies beginning in infancy could chart the developmental trajectories of parent-child neural synchrony and their impact on long-term social outcomes. Such work could also explore sensitive periods when parenting may be most influential on social brain maturation. Finally, expanding this multimodal approach to clinical populations like autism could yield insights into atypical social cognitive development and inform tailored intervention strategies targeting parent-child relationships and neural plasticity.

## Conclusions

Our study provides novel evidence that children's social cognitive development may be shaped by the intricate interplay between environmental influences, such as parenting, and biological factors, such as neural maturation. Our findings contribute to a growing understanding of the factors associated with social cognitive development and suggest the potential importance of parenting in this process. Specifically, the study points to the possible role of the parent-child relationship in supporting the development of social brain circuitry and highlights the relevance of family-based approaches for addressing social difficulties. The observed neural synchronization between parent and child, which was associated with relationship quality, underscores the potential significance of positive parental engagement in fostering social cognitive skills. Future longitudinal and clinical research can build on this multimodal approach to further clarify the neurobehavioral mechanisms underlying social cognitive development. Such research may help inform more effective strategies

for promoting healthy social functioning and mitigating social deficits through targeted family-based interventions.

## Methods

### Participants

A total of 50 child-mother dyads (100 participants) were recruited for inclusion in the study from local communities via flyers or internet advertisements. The children had a mean age of 5.43 years (SD = ±1.75 years, range = 3–8 years), and the adults had a mean age of 36.7 years (SD = ±4.0 years, range = 30.7–48.8 years). We chose to focus exclusively on mother-child dyads in this study based on prior evidence suggesting distinct neural and behavioral caregiving profiles between mothers and fathers (*Swain et al., 2014*; *Abraham et al., 2014*), allowing us to maintain role consistency and reduce variability in dyadic interactions. Following quality control, 16 dyads were excluded due to excessive head motion during fMRI scanning, resulting in a final sample of 34 dyads for analysis. None of the participants had a history of psychiatric or neurological disorders, and all had normal or corrected-to-normal visual acuity. Written informed consent was obtained from all participants after a thorough explanation of the study's purpose and procedures. For child participants, written parental consent and age-appropriate child assent were also obtained. The study was approved by the Ethics Committee of the University of Electronic Science and Technology of China (approval number: 1061423031025276) and conducted in accordance with the principles of the Declaration of Helsinki. All data were anonymized before analysis to protect participant privacy.

### Questionnaires

Parents or guardians completed several questionnaires to assess various aspects of the child-parent relationship and the child's social behavior:

1. Child-Parent Relationship Scale (CPRS): This 22-item scale evaluates the child-mother relationship through two subscales: closeness (10 items) and conflict (12 items). Higher scores indicate higher levels of closeness or conflict. Responses range from 1 (definitely does not apply) to 5 (definitely applies). This scale has demonstrated good psychometric properties in Chinese participants (*Pianta, 1992*).
2. Egna Minnen Betriiffande Uppfostran (EMBU): This scale assesses parents' own parenting behavior toward their child. It includes four subscales: rejection (13 items), emotional warmth (17 items), control attempts (19 items), and preference (3 items). Higher scores indicate greater levels of each respective behavior. Responses range from 1 (no, never) to 4 (yes, always) (*Zhang, 2008*).
3. Social Responsiveness Scale (SRS): This 65-item rating scale evaluates the child's social behavior over the past 6 months, generating a total score that indicates the severity of social deficits. Higher scores reflect greater social impairment. Responses range from 0 (no, never true) to 3 (almost always true) (*Constantino and Gruber, 2012*).

### Neurodevelopmental assessment

The Peabody Picture Vocabulary Test, Fourth Edition (PPVT-4), was used to assess the receptive vocabulary abilities of the children (*Dunn and Dunn, 1965*), providing an estimate of verbal intelligence (*Dunn and Dunn, 1981*). The Chinese version of the PPVT-4 has been adapted to align closely with the original version, with only minor adjustments for cultural and language differences. Each correct item was awarded one point, and raw scores were converted to standardized scores according to PPVT-4 protocols.

#### Explicit ToM task and false-belief composite score

All children participated in a custom-made explicit ToM battery, which entailed listening to a series of narrative scenarios and subsequently responding to questions that required them to infer the mental states of the characters involved (Appendix 1). This battery was specifically designed to assess first-order and second-order ToM abilities, and has been widely used in previous neuroimaging studies to evaluate children's ToM behavior performance (*Sullivan et al., 1994*; *Hughes et al., 2000*). As described in previous literatures (*Richardson et al., 2018*; *Jones et al., 2010*), first-order ToM is

associated with the ability to infer other's mental state (e.g. "where does the girl think her ball is?"), and second-order TOM is associated with the ability to infer other's mental state in relation to a third party (e.g. "where does the girl think the boy will go looking for his ball?"). Each task includes a target (e.g. "where does the girl think her ball is?") and a control question (e.g. "where is the actual location of the ball?"), and children received one point for each correct answer to both the test and control questions. A total of 14 tasks were assigned, and the performance was quantified as the total number of correct answers out of 14 tasks, resulting in the performance scores ranging from 0 to 14.

## fMRI paradigm and stimuli

Participants watched '*Partly Cloudy,*' a 5.6 min animated movies (*Reher and Sohn, 2009*; https://www.pixar.com/partly-cloudy#partly-cloudy-1), which was originally adopted and validated by Jacoby and Richardson et al. to elicit complex social-cognitive reasoning in an fMRI setting (*Jacoby et al., 2016*; *Richardson et al., 2018*; *Richardson et al., 2020*). Before the movie commenced, participants were given a 10 s rest period to stabilize their attention and physiological state. During this time, they were instructed to relax, remain still, and prepare for the upcoming task. To ensure precise synchronization between the film's presentation and the neuroimaging data acquisition, both the stimuli and the scanning process were initiated simultaneously, allowing for accurate time-point alignment from the start of the experiment.

## fMRI data acquisition

Participants were scanned using a 3T GE DISCOVERY MR750 scanner (General Electric), equipped with an eight-channel prototype quadrature birdcage head coil. Functional images were acquired using an echo-planar imaging (EPI) sequence with the following parameters: echo time (TE) of 30 ms, repetition time (TR) of 2000 ms, and a spatial resolution of 3.75×3.75×3.2 mm. The imaging matrix was 64×64, with a flip angle of 90°, and a field of view (FOV) of 240×240 mm². A total of 43 interleaved transverse slices, with no inter-slice gap and a slice thickness of 3.2 mm, were obtained. The functional dataset comprised 155 dynamic scans, providing comprehensive coverage of brain activity during the task.

## fMRI data preprocessing

As previously published (*Li et al., 2022*; *Yan et al., 2016*), the standard fMRI preprocessing pipeline was applied using the Data Processing and Analysis of Brain Imaging (DPABI) toolbox, version 4.3 (https://rfmri.org/DPABI). To ensure data quality, the first five volumes from each participant were discarded due to initial signal instability. The remaining 145 volumes underwent slice-timing correction and head-motion realignment. Participants were excluded if either the child or the mother exhibited excessive head motion, defined as translation exceeding 3.0 mm or rotation greater than 3°. Consequently, 16 child-mother dyads were excluded, leaving a total of 34 dyads for analysis. The brain images of all remaining children and mothers were spatially normalized to the Montreal Neurological Institute (MNI) standard template and resampled to a 3×3×3 mm³ voxel size. This normalization allowed for the application of group ROIs and hypothesis spaces derived from adult datasets, facilitating direct comparisons between child and mother participants. Previous research supports the use of a common space for both adults and children under age seven, as anatomical differences in this age group are minimal relative to the fMRI resolution (*Burgund et al., 2002*; *Cantlon et al., 2006*). Subsequently, the normalized images were linearly detrended to mitigate signal drifts. Nuisance covariates, including 24 head-motion parameters, white matter signal, and cerebrospinal fluid signal, were regressed out of the data (*Friston et al., 1996*). To enhance signal-to-noise ratio, all images were smoothed with a 6×6×6 mm³ full-width at half-maximum Gaussian kernel. A bandpass filter (0.01–0.15 Hz) was applied to isolate the neural signal associated with movie viewing (*Meer et al., 2020*). Additionally, data scrubbing was performed to address potential motion artifacts (*Power et al., 2012*). Signal outliers, identified as having a framewise displacement (FD) greater than 0.5 mm with the preceding 1 and subsequent 2 volumes, were corrected by fitting these outliers to the clean portion of the time series using a third-order spline.

## Behavior analysis

Of the 50 initial mother-child dyads recruited, 16 were excluded due to excessive head motion (n=11), incomplete scan sessions (n=3), or technical issues during data acquisition (n=2). The final sample consisted of 34 dyads. To assess potential bias introduced by data exclusion, we compared included and excluded dyads on child age, gender, and Theory of Mind performance. Comparison between included and excluded dyads revealed no significant differences in child age (t=1.23, $p$=0.24), ToM scores (t=–0.54, $p$=0.59), or sex distribution ($\chi^2$<0.01, $p$=0.98), indicating that data exclusion did not bias the sample in a systematic way. No significant differences were found across these variables (all $ps$>0.1), suggesting that the analytic sample was demographically representative of the full cohort. Multiple linear regression and Pearson correlation analysis was performed to evaluate the relationship between ToM behavior and general demographic and behavioral measures.

## Reverse correlation analysis

Timecourse analyses were performed by extracting preprocessed timecourses from 9 mm ROIs centered on peaks identified in previous studies (6 ToM regions, 7 SPM regions, *Table 2*; *Richardson et al., 2018*). Each ROI timecourse was z-normalized to standardize the data. For each network, the timecourses across ROIs were averaged, resulting in a single timecourse for the ToM network and another for the SPM network per participant. We employed reverse correlation analysis in adults to identify discrete events within the movie that elicited reliable neural responses across participants in ToM and SPM networks (*Richardson et al., 2018*). The network timecourse for each timepoint across adult subjects was compared to the baseline (0) using a one-tailed *t*-test. Events were defined as sequences of two or more consecutive timepoints with significantly positive responses within each network. Since reverse correlation analyses assess responses relative to baseline, the identified events are considered to drive responses in the ROIs compared to other moments in the stimulus. Events were then rank-ordered based on the average magnitude of response at the peak timepoint and labeled accordingly (e.g. event T01 represents the ToM event that elicited the highest magnitude of response in the ToM network). The events of adults were chosen for this analysis due to the relative stability and maturity of their social brain responses, allowing for robust detection of canonical ToM and social pain-related moments. These events, once identified, served as stimulus-locked timepoints for subsequent analyses in the child cohort. This approach enables us to examine how children's responses to well-characterized, socially meaningful events vary with age and parent-child dyadic dynamics.

## Inter-region correlation analyses and functional neural maturity

In inter-region correlation analyses, each ROI timecourse was correlated with every other ROI's timecourse to obtain correlation matrices, per subject, and these correlation values were Fisher z-transformed. Then, the average correlation matrices were conducted for different age groups. Within-ToM correlations were the average correlation from each ToM ROI to every other ToM ROI, within- SPM correlations were the average correlation from each SPM ROI to every other SPM ROI, and across-network correlations were the average correlation from each ToM ROI to each SPM ROI. Pearson correlation was used to determine the relationship between age and the within-ToM/SPM network correlations.

Next, we tested whether the functional neural maturity of each child's timecourse responses (i.e. similarity to adults) was related to the inter-region network correlations. Neural maturity was calculated as average value of the Pearson correlation between child's correlation matrices and each adult's correlation matrices. Pearson correlation was used to determine the relationship between neural maturity and age/relationship quality. Pair *t*-tests were performed on the neural maturity during ToM/SP events and other events between different age groups. In addition, a 2×3 ANOVA with neural maturity as the dependent variable was used to determine the effects of ToM/SP events (ToM/SP events vs. other events), development (Pre-junior group vs. Junior group, Senior group), and their interaction.

## ISS in fMRI time series

ISS was defined as the temporal correlation between the mean timecourses of social brain networks (averaged across ROIs) in child-mother dyads, determined through Pearson correlation analysis.

Similarly, ISS for child-stranger dyads was calculated by correlating each child's network timecourse with the average adult timecourse (excluding their own mothers). The Fisher's r-to-z transform was applied to convert the correlation coefficient (r) values to normally distributed z scores. Two-sample t-tests were then conducted to compare ISS values between child-mother dyads and child-stranger dyads. Partial correlation analysis, with children's age as a covariate, was performed separately for child-mother and child-stranger dyads to examine the relationship between ISS and relationship quality.

A 2×2 ANOVA was used with ISS as the dependent variable to assess the effects of ToM/Pain events (ToM/SP events vs. other events), parental relationship (child-mother dyads vs. child-stranger dyads), and their interaction. Pairwise *t*-tests were conducted to compare ISS values between child-mother and child-stranger dyads across ToM/SP events and other events.

### Structural equation modeling

SEM was conducted using R package lavaan (*Rosseel, 2012*). The analyses employed lavaan settings consistent with standard practices in other software packages like AMOS. These settings included Wishart estimation, maximum likelihood estimation for handling missing data, and the use of expected information for estimating standard error variance. Within the hypothetical model, a latent factor was constructed for a *personal growth/trait* variable, another latent factor was constructed for a *parenting* variable, and social responsiveness was represented as a manifest variable for output. *Personal growth/ trait* and ToM behavior were included in the regression as the mediator. The hypothesized research model is depicted in *Figure 5*. To determine the model fit, we examined the $\chi^2$/df ratio, Comparative Fit Index (CFI), and root mean square error of approximation (RMSEA). A good model fit is reflected by $\chi^2$/df ratios <3 (*Kline, 2023*), fit indices above 0.90 (*Bentler, 1990*), and RMSEA values ≤0.10.

### Code availability

All process of code needed to evaluate the conclusions in the paper are present in the paper and/or the Appendix. Additionally, this study used openly available software and codes, specifically DPABI (https://rfmri.org/DPABI) and R Package lavaan (https://cran.r-project.org/web/packages/lavaan/lavaan.pdf).

### Acknowledgements

This study was supported by the National Natural Science Foundation of China (82322035, 62273076, 82121003, and 62036003), Fundamental Research Funds for Central Universities (ZYGX2019Z017), and National Social Science Foundation of China (20&ZD296).

## Additional information

#### Funding

| Funder | Grant reference number | Author |
| --- | --- | --- |
| National Natural Science Foundation of China | 82322035 | Xujun Duan |
| Fundamental Research Funds for the Central Universities | ZYGX2019Z017 | Xujun Duan |
| National Social Science Fund of China | 20&ZD296 | Xujun Duan |
| National Natural Science Foundation of China | 62273076 | Xujun Duan |
| National Natural Science Foundation of China | 82121003 | Xujun Duan |
| National Natural Science Foundation of China | 62036003 | Xujun Duan |

| Funder | Grant reference number | Author |
|--------|------------------------|--------|

The funders had no role in study design, data collection and interpretation, or the decision to submit the work for publication.

## Author contributions

Lei Li, Conceptualization, Data curation, Software, Formal analysis, Validation, Investigation, Visualization, Methodology, Writing - original draft, Writing – review and editing; Jinming Xiao, Data curation, Software, Validation, Investigation, Methodology; Weixing Zhao, Qingyu Zheng, Xinyue Huang, Xiaolong Shan, Data curation, Methodology; Yating Ming, Peng Wang, Data curation; Zhen Wu, Conceptualization, Investigation, Methodology; Huafu Chen, Resources, Supervision, Funding acquisition, Investigation, Project administration; Vinod Menon, Supervision, Investigation, Methodology, Writing - original draft, Writing – review and editing; Xujun Duan, Conceptualization, Resources, Supervision, Funding acquisition, Project administration, Writing – review and editing

## Author ORCIDs

Xujun Duan ⓘD https://orcid.org/0000-0001-8543-2117

## Ethics

Human subjects: Written informed consent was obtained from all participants after a thorough explanation of the study's purpose and procedures. For child participants, written parental consent and age-appropriate child assent were also obtained. The study was approved by the Ethics Committee of the University of Electronic Science and Technology of China (approval number: 1061423031025276) and conducted in accordance with the principles of the Declaration of Helsinki. All data were anonymized before analysis to protect participant privacy.

Reviewer #1 (Public review): https://doi.org/10.7554/eLife.103017.3.sa1
Reviewer #2 (Public review): https://doi.org/10.7554/eLife.103017.3.sa2
Reviewer #3 (Public review): https://doi.org/10.7554/eLife.103017.3.sa3
Author response https://doi.org/10.7554/eLife.103017.3.sa4

# Additional files

## Supplementary files
MDAR checklist

## Data availability

Anonymized fMRI used for analyses have been deposited in a Zenodo under "Neurobehavioral Synchrony and Social Cognition Development" (https://zenodo.org/records/12730121).

The following dataset was generated:

| Author(s) | Year | Dataset title | Dataset URL | Database and Identifier |
|-----------|------|---------------|-------------|-------------------------|
| Li L, Xiao J, Zhao W, Zheng Q, Huang X, Shan X, Ming Y, Wang P, Wu Z, Chen H, Menon V, Duan X | 2022 | Neurobehavioral Synchrony and Social Cognition Development | https://doi.org/10.5281/zenodo.12730121 | Zenodo, 10.5281/zenodo.12730121 |

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

## Appendix 1

### Theory of Mind (ToM) behavioral tasks
Overview
Children completed a structured battery of 14 Theory of Mind (ToM) questions, designed to assess multiple components of social-cognitive reasoning. These tasks were adapted from prior validated paradigms and covered a range of developmental levels, including false belief understanding, knowledge access, deception, and second-order reasoning. Tasks were administered in a fixed order from simple to complex. Each task included a brief narrative and one or more questions targeting the child's ability to infer mental states (e.g. beliefs, intentions, perspectives). Correct responses to test questions (with or without control questions) were scored as 1; incorrect or no response was scored as 0. The total score ranged from 0 to 14.

To reduce cognitive fatigue for younger children, testing was discontinued if a child failed two consecutive tasks, reflecting a ceiling in their performance level.

### Task descriptions
Note: In the following descriptions,

- 🟧 = experimenter's spoken instructions
- 🟦 = experimenter's actions
- 🟩 = story narration

#### Tasks 1–2: Unexpected contents task (rock in candy wrapper)

- 🟧 "Look, what do I have in my hand?"
- 🟩 (The object is a wrapped candy.)
- 🟧 "What do you think is inside?" → "Candy."
- 🟧 "Great, now let's open it and see what's really inside."
- 🟦 (The experimenter unwraps the candy in front of the child.)
- 🟧 "What's actually inside?" → "A rock."
- 🟦 (The experimenter re-wraps the candy.)
- 🟧 Control Question: "Do you know what's inside the wrapper now?" → "A rock."
- 🟧 Test Question 1: "Before we opened it, what did you think was inside?" → "Candy."
- 🟧 Test Question 2: "Now another child walks in and hasn't seen what's inside. What will they think is in the wrapper?" → "Candy."

#### Task 3–4: Unexpected location task (ping-pong ball in colored cups)

- 🟧 "What color are these two cups?" → "Blue and red."
- 🟩 "Now here comes a boy and a girl. They are playing with a ball together."
- 🟦 (Experimenter uses figures to act out playing)
- 🟩 "The boy gets hungry and goes to eat. To hide the ball, he puts it in the blue cup and leaves."
- 🟩 "The girl, not hungry yet, takes the ball out and plays some more."
- 🟦 (Experimenter acts out girl playing)
- 🟩 "Then she hides the ball in the red cup before going to eat."
- 🟧 Memory Questions:

  – "Where did the boy hide the ball at first?" → "Blue cup"
  – "Did the boy see the girl move the ball?" → "No"
- 🟧 Test Question 3: "When the boy comes back, where will he think the ball is?" → "Blue cup"
- 🟧 Test Question 4: "Which cup will he look in first?" → "Blue cup

## Tasks 5–8: Birthday bunny scenario

■ "Today is Xiaoming's birthday. He really wants a bunny. His mom actually bought him the bunny he wanted most, but she wanted to surprise him. So she hid the bunny in the cupboard and didn't tell Xiaoming."

■ (Experimenter says as the mom): "Sorry Xiaoming, I didn't buy you a bunny for your birthday. I got you an electric fan instead."

■ Control Question 1: "Did Xiaoming's mom really get him an electric fan for his birthday?" → "No"

■ Control Question 2: "Did the mom tell Xiaoming she got him an electric fan?" → "Yes"

■ Control Question 3: "Why did the mom say she got an electric fan?" → "To surprise him."

■ "Then, the mom went out. Xiaoming went to the cupboard to get some clothes. When he opened it, he discovered the bunny! He was so happy and said: 'Wow! Mom really got me the bunny I wanted!'"

■ "Remember: the mom didn't see Xiaoming go to the cupboard or find the bunny."

■ Test Question 5: "Does Xiaoming now know that his mom got him a bunny for his birthday?" → "Yes"

■ Test Question 6: "Does the mom know that Xiaoming found the bunny?" → "No"

■ "Later, Xiaoming left with his bunny. Then, the mom returned home with Xiaoming's grandmother."

■ Test Question 7: (As Grandma asks the mom): "Does Xiaoming know what gift you really got him for his birthday?'"

■ "What does the mom think — does Xiaoming know, or not?" → "Not"

■ Memory cue: "Remember, the mom didn't see Xiaoming find the bunny."

■ Test Question 8: (Grandma continues): "Then what does Xiaoming think the gift is?"

■ "What does the mom think Xiaoming believes the gift is — the bunny, or the electric fan?" → "Electric fan"

■ Test Question 9: "Why does the mom think so?" → "Because she told him it was a fan, and doesn't know he saw the bunny."

## Tasks 9–12: Picture book deception scenario

■ "A girl had a picture book she really liked. One day, a boy came over to her house to play. He said: 'I heard you have a really fun picture book! Where did you put it? I want to read it too.'"

■ "But the girl was afraid the boy might damage the book, so she didn't want to share it. She decided to lie and said: 'Sorry, I left my picture book at kindergarten. I didn't bring it home, so you can't read it.'"

■ Control Question 1: "Was the picture book actually left at kindergarten?" → "No"

■ Control Question 2: "Did the girl tell the boy the book was at kindergarten?" → "Yes"

■ Control Question 3: "Why did the girl say that?" → "Because she was afraid he'd damage it and wanted to keep it for herself."

■ "Later, the boy got bored and went out to play. As soon as he left, the girl was happy she could finally read her book alone. She went to the cupboard to get it."

■ (As the girl opens the cupboard and takes out the picture book)

■ "At that very moment, the boy turned back and saw her taking the book from the cupboard."

■ "But the girl didn't notice that the boy had seen her."

■ Test Question 10: "Does the boy now know where the book really is?" → "Yes"

■ Test Question 11: "Does the girl know the boy saw her get the book?" → "No"

■ "Later, the girl's mom came home and asked:"

■ Test Question 12: "'Does the boy know where the book is?'"

■ "What does the girl think — does the boy know or not?" → "Not"

■ Memory cue: "Remember, the girl didn't see the boy watching her take the book from the cupboard."

■ Test Question 13: "What does the girl think the boy believes — where is the book?" → "At kindergarten"

- Test Question 14: "Why does she think that?" → "Because she told him it was there and didn't see him watching her."

## Appendix 2

### Additional analyses and results

To further investigate the specificity of our findings, we conducted additional control analyses focusing on the individual components of the social brain networks examined in our study: the ToM and SPM networks.

When analyzing these networks separately, we found significant correlations between neural maturity and age, as well as between ISS and parent-child relationship quality for both the ToM and SPM networks individually (*Appendix 2—figure 1*). Specifically, neural maturity within each network was positively correlated with age, indicating that both networks undergo maturation during childhood. Similarly, ISS within each network was negatively correlated with parent-child conflict scores, suggesting that both networks contribute to the observed relationship between neural synchrony and parent-child relationship quality.

However, when we applied our structural equation model to each network separately, we found that the interactive effect of parenting and personal growth factors could not significantly predict children's social cognition outcomes (*Appendix 2—figure 2*). This finding suggests that the integration of information from both the ToM and SPM networks is essential for comprehensively predicting social cognitive outcomes in children.

These results highlight the importance of considering the social brain as an integrated system, where the ToM and SPM networks work in concert to support social cognitive development. While each network shows age-related maturation and sensitivity to parent-child relationship quality, their combined functioning appears to be crucial for predicting broader social cognitive outcomes.

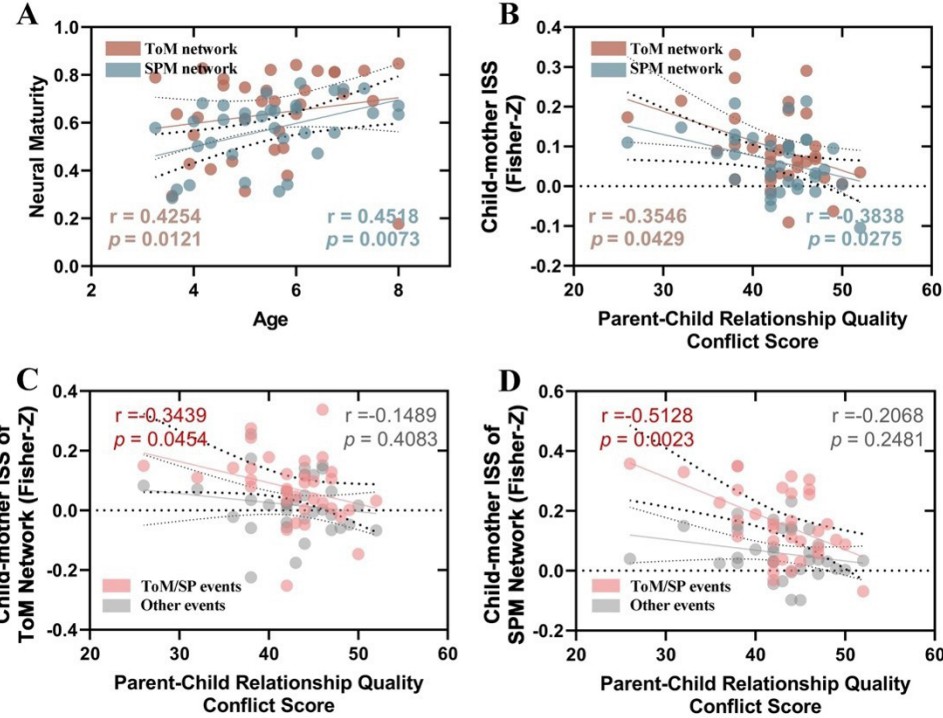

**Appendix 2—figure 1.** Relationship between personal traits, neural measures and parent-child relationship quality for Theory of Mind (ToM) and Social Pain Matrix (SPM) networks analyzed separately. (**A**) Correlation between neural maturity within ToM and SPM networks and age. (**B**) Correlation between parent-child relationship quality and inter-subject synchronization (ISS) within ToM and SPM networks in child-mother dyads. (**C–D**) Partial correlation between Child-Parent Relationship Scale (CPRS) conflict scores and ISS within ToM/SPM networks in child-mother dyads during ToM/Social Pain and other events, shown in red and black circles, respectively.

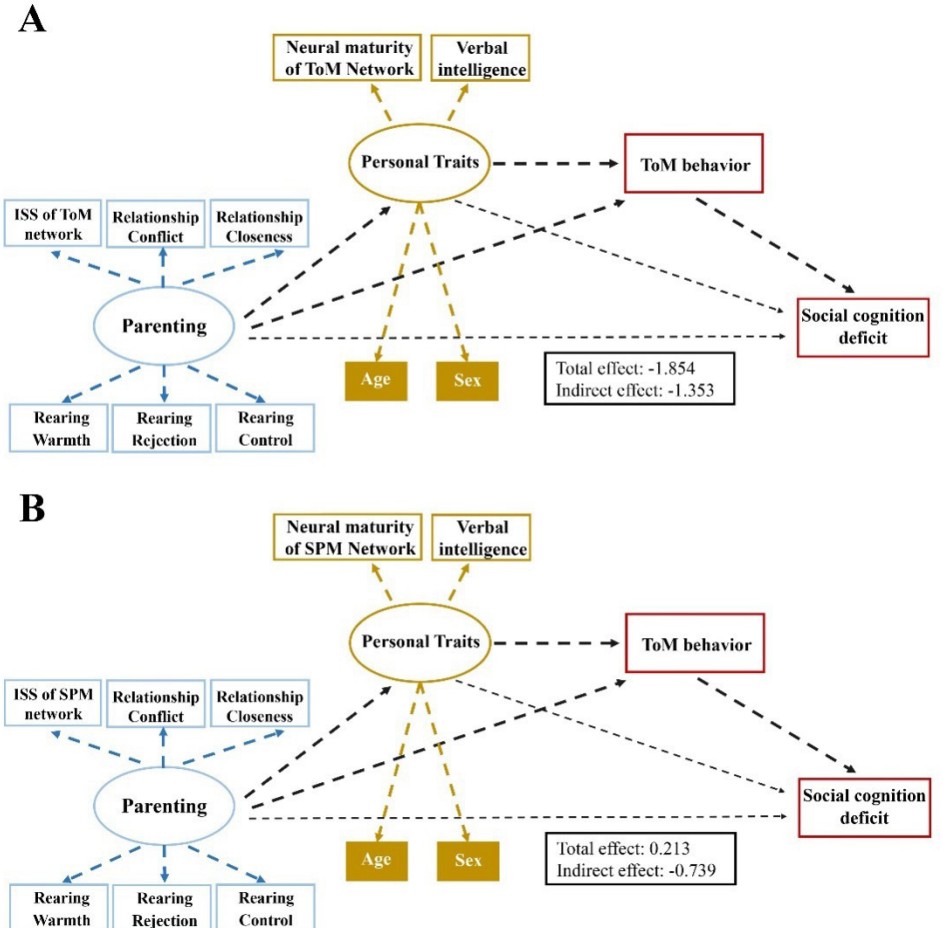

**Appendix 2—figure 2.** Structural equation modeling using (**A**) Theory of Mind (ToM) and (**B**) Social Pain Matrix (SPM) networks separately. Solid lines indicate significant paths, and dashed arrows indicate nonsignificant paths. No significant direct effects on social cognition outcomes were observed when analyzing the networks individually, highlighting the importance of considering both networks together for predicting social cognitive outcomes.

**Appendix 2—table 1.** Theory of Mind (ToM) and Social Pain event details.

Time (Events in neural responses were then shifted 6 s in time to account for the hemodynamic lag), duration (seconds), and description for each ToM and Social Pain event. Event labels (T01, P01) reflect rank order of average response magnitude in adults.

**ToM events**

| Events | Time point (in stimulus) | Duration | Description |
|---|---|---|---|
| T07 | 1:00-1:04 | 4s | Clouds create baby animals accompanied by laughter. |
| T02 | 1:12-1:18 | 6s | Baby cries, Cloud gives him a helmet, and he's happy. |
| T04 | 1:30-1:36 | 6s | Gus is a lonely cloud. |
| T01 | 2:38-2:50 | 12 s | Peck (crane) gazed longingly at the happy cloud that was making puppies and wore a look of envy. |
| T06 | 2:54-2:58 | 4s | Peck notices that Gus caught him looking longingly and feels bashful. |
| T05 | 3:18-3:22 | 4s | Gus saw Peck scream in agony as he was stabbed by the porcupine's thorns and exclaimed. |
| T03 | 3:34-4:56 | 22 s | Peck dons football gear to explain to Gus that he did not abandon him, but rather was acquiring protective equipment so that he could continue to deliver Gus's babies. |

*Appendix 2—table 1 Continued on next page*

*Appendix 2—table 1 Continued*

**ToM events**

### Social Pain events

| Events | Time point (in stimulus) | Duration | Description |
|---|---|---|---|
| P07 | 0:48-0:54 | 6s | Cloud makes baby cat (lightning). |
| P06 | 1:08-1:12 | 4s | Puppy's chewing on a bone. |
| P08 | 1:24-1:32 | 8s | Gus makes baby alligator (lightning). |
| P05 | 2:04-2:08 | 4s | Peck got bitten on the head by a baby alligator. |
| P03 | 3:02-3:14 | 12 s | Peck's wing was repeatedly stabbed by porcupine baby's pinprick |
| P01 | 3:24-3:28 | 4s | Gus pulls porcupine spines out of Peck's head |
| P04 | 4:12-4:18 | 6s | Gus expresses anger through thunder |
| P02 | 4:52-5:00 | 8s | Peck is electrocuted by baby eel (lightning). |

**Appendix 2—table 2.** Structural equation modeling (SEM) results.

| X→ | Y | Estimate | St. Err | Z-value | p |
|---|---|---|---|---|---|
| **Latent Variables** | | | | | |
| Neural maturity | Personal growth | 1.00 | | | |
| Verbal intelligence | Personal growth | 0.452 | 0.290 | −1.556 | 0.12 |
| Age | Personal growth | 1.243 | 0.560 | 2.238 | 0.025 |
| Sex | Personal growth | −1.234 | 0.552 | −2.236 | 0.025 |
| ISS | Parenting | 1.00 | | | |
| Relationship conflict | Parenting | −0.797 | 0.261 | −3.047 | 0.002 |
| Relationship closeness | Parenting | 0.500 | 2.254 | 1.971 | 0.049 |
| Rearing warmth | Parenting | −0.174 | 0.249 | −0.698 | 0.485 |
| Rearing rejection | Parenting | 0.989 | 0.268 | 3.692 | <0.001 |
| Rearing control | Parenting | −0.619 | 0.257 | −2.412 | 0.016 |
| **Regressions** | | | | | |
| Parenting | Personal growth | −0.708 | 0.271 | −2.618 | 0.009 |
| Parenting | ToM behavior | 0.550 | 0.288 | 1.909 | 0.050 |
| Personal growth | ToM behavior | −0.647 | 0.338 | −1.916 | 0.049 |
| ToM behavior | Social cognition deficit | −0.458 | 0.209 | −2.193 | 0.028 |
| Parenting | Social cognition deficit | 0.427 | 0.319 | −1.336 | 0.182 |
| Personal growth | Social cognition deficit | −0.491 | 0.323 | −1.521 | 0.128 |
| **Defined parameters:** | | | | | |
| Direct effect | | −0.491 | 0.323 | −1.521 | 0.128 |
| Indirect effect | | −0.210 | 0.106 | −1.976 | 0.048 |
| Total | | −0.651 | 0.325 | −2.005 | 0.045 |

