## [Editor Report · eLife Assessment]

This **important** study reports **solid** evidence for the significant role of mother-child neural synchronization and relationship quality in the development of Theory of Mind (ToM) and social cognition. The findings effectively bridge brain development with children's behavior and parenting practices, and will be of interest to researchers studying brain development and social cognition, as well as the general public.

---

## [Referee Report · Reviewer #1 (Public review)]

The authors have undertaken a significant revision of the manuscript and addressed the vast majority of our original comments. The manuscript is significantly improved as a result and will make a nice contribution to the literature. The new framing is especially impactful.

We have a few remaining comments to improving the manuscript:

Q1: The authors clarified the multiple comparison correction appropriately, and included a comprehensive of the study limitations related to causality and SEM. We think there could be a few further improvements to the manuscript to fully address our initial comment.

Under the results section where the authors describe the use of structural equation modeling, we think that it would be helpful to readers to further emphasize that the current design doesn't allow for delineation of temporal sequences in development and do cannot reflect true mediation. These are important caveats that the readers describe beautifully in their response.

In addition to think about the mediating variables, can the authors conduct a sensitivity analysis that re-orders the IV, mediator, and DV? That way, a formal comparison can be made between model fits. It would provide an empirical basis for how to temper the discussion of these findings.

Q7: We think that this analysis (lack of significant correlations between ISS, child age, and neural maturity) and corresponding discussion by the authors would be very interesting for readers. It does not appear as though they've added this information to the text (even in a supplementary file would suffice), but I think their conclusions about the data are strengthened related to context specific neural dynamics.

---

## [Referee Report · Reviewer #2 (Public review)]

Summary:

This study investigates the impact of mother-child neural synchronization and the quality of parent-child relationships on the development of Theory of Mind (ToM) and social cognition. Utilizing a naturalistic fMRI movie-viewing paradigm, the authors analyzed inter-subject neural synchronization in mother-child dyads and explored the connections between neural maturity, parental caregiving, and social cognitive outcomes. The findings indicate age-related maturation in ToM and social pain networks, emphasizing the importance of dyadic interactions in shaping ToM performance and social skills, thereby enhancing our understanding of the environmental and intrinsic influences on social cognition.

Strengths:

This research addresses a significant question in developmental neuroscience, by linking social brain development with children's behaviors and parenting. It also uses a robust methodology by incorporating neural synchrony measures, naturalistic stimuli, and a substantial sample of mother-child dyads to enhance its ecological validity. Furthermore, the SEM approach provides a nuanced understanding of the developmental pathways associated with Theory of Mind (ToM). The manuscript also addressed many concerns raised in the initial review. The adoption of the neuroconstructivist framework effectively frames neural and cognitive development as reciprocal, addressing prior concerns about causality. The justification for methodological choices, such as omitting resting-state baselines due to scanning challenges in children and using unit-weighted scoring for ToM tasks, further strengthens the study's credibility.

Weaknesses:

(1) The revised introduction has improved, particularly in framing the first goal-developmental changes in ToM and SPM networks-as a "developmental anchor" for goals 2 and 3. However, given prior research on age-related changes in these networks (e.g., Richardson et al., 2018), the authors should clarify whether this goal seeks to replicate prior findings or to extend them under new contexts. Specifying how this part differs from existing work and articulating specific hypotheses would enhance the focus.

(2) I still have some reservations about retaining the slightly causal term "shape" in the title. While the manuscript now carefully avoids causal claims, the title may still be interpreted as implying directionality, especially by non-specialist audiences.

(3) One more question about Figure 2A and 2B: adults and children showed highly similar response curves for video frames, yet some peaks (e.g., T02, T05, T06) are identified as ToM or SPM events only in adults. Whether statistical methods account for the differences? Or whether the corresponding video frames contain subtle social cues that only adults can process?

---

## [Referee Report · Reviewer #3 (Public review)]

Summary:

The article explores the role of mother-child interactions in the development of children's social cognition, focusing on Theory of Mind (ToM) and Social Pain Matrix (SPM) networks. Using a naturalistic fMRI paradigm involving movie viewing, the study examines relationships among children's neural development, mother-child neural synchronization, and interaction quality. The authors identified a developmental pattern in these networks, showing that they become more functionally distinct with age. Additionally, they found stronger neural synchronization between child-mother pairs compared to child-stranger pairs, with this synchronization and neural maturation of the networks associated with the mother-child relationship and parenting quality.

Strengths:

This is a well-written paper, and using dyadic fMRI and naturalistic stimuli enhances its ecological validity, providing valuable insights into the dynamic interplay between brain development and social interactions.

Weaknesses:

The current sample size (N = 34 dyads) is a limitation, particularly given the use of SEM, which generally requires larger samples for stable results. Although the model fit appears adequate, this does not guarantee reliability with the current sample size.

---

## [Author Response]

The following is the authors’ response to the original reviews

**Reviewer #1:**
The authors sought to examine the associations between child age, reports of parent-child relationship quality, and neural activity patterns while children (and also their parents) watched a movie clip. Major methodological strengths include the sample of 3-8 year-old children in China (rare in fMRI research for both age range and non-Western samples), use of a movie clip previously demonstrated to capture theory of mind constructs at the neural level, measurement of caregiver-child neural synchrony, and assessment of neural maturity. Results provide important new information about parent-child neural synchronization during this movie and associations with reports of parent-child relationship quality. The work is a notable advance in understanding the link between the caregiving context and the neural construction of theory of mind networks in the developing brain.

We are grateful for the reviewer’s generous and thoughtful summary of our work. We particularly appreciate the recognition of the methodological strengths—including the rare developmental sample, culturally diverse context, and use of naturalistic, theory of mind-relevant stimuli—as well as the importance of integrating neural synchrony and relational variables. The reviewer’s comments affirm the core motivation behind this study: to advance our understanding of how the caregiving environment shapes the neurodevelopment of social cognition in early childhood. We have taken all specific suggestions seriously and hope the revised manuscript more clearly communicates these contributions.

We appreciate that the authors wanted to show support for a mediational mechanism. However, we suggest that the authors drop the structural equation modeling because the data are cross-sectional so mediation is not appropriate. Other issues include the weak justification of including the parent-child neural synchronization as part of parenting.... it could just as easily be a mechanism of change or driven by the child rather than a component of parenting behavior. The paper would be strengthened by looking at associations between selected variables of interest that are MOST relevant to the imaging task in a regression type of model. Furthermore, the authors need to be more explicit about corrections for multiple comparisons throughout the manuscript; some of the associations are fairly weak so claims may need to be tempered if they don't survive correction.

Thanks for feedback on the use of SEM in our study. We recognize the limitations of using SEM to infer mediation with cross-sectional data and acknowledge that longitudinal designs are better suited for such analyses. However, our goal was not to establish causality but to explore potential pathways linking parenting, personal traits, and Theory of Mind (ToM) behavior to social cognition outcomes. SEM allowed us to simultaneously examine the relationships among these latent constructs, providing a cohesive framework for understanding the interplay of these factors. That said, we understand your concern and are willing to revise the manuscript to de-emphasize causal interpretations of the SEM findings.

We thank the reviewer for raising the corrections for multiple comparisons. We confirm that all correlation analyses reported in the manuscript have been corrected for multiple comparisons using the False Discovery Rate (FDR) procedure. In the revised manuscript, we now explicitly indicate FDR correction for all relevant p-values to ensure clarity and transparency. Where this information was previously missing, we have corrected the oversight and clearly labeled the results as FDR-corrected or uncorrected where appropriate. Additionally, we have carefully reviewed our interpretation of all reported associations. For any results that were close to the significance threshold, we have tempered our claims and now describe them as a marginally significant association to avoid overstating our findings.

The corresponding changes have been made on Discussion section of the revised manuscript.

Reverse correlation analysis is sensible given what prior developmental fMRI studies have done. But reverse correlation analysis may be more prone to overfitting and noise, and lacks sensitivity to multivariate patterns. Might inter-subject correlation be useful for *within* the child group? This would minimize noise and allow for non-linear patterns to emerge.

We appreciate the reviewer’s thoughtful suggestion regarding potential limitations of reverse correlation analysis. While we agree that inter-subject correlation (ISC) within the child group may be useful in other contexts, our primary goal in using reverse correlation was not to identify temporally distributed or multivariate response patterns, but rather to isolate specific events within the naturalistic stimulus that reliably evoke Theory of Mind (ToM) and Social Pain-related responses in adults—who possess more stable and mature neural signatures. These adult-derived events serve as anchors for subsequent developmental comparisons and provide a principled way to define timepoints of interest that are behaviorally and theoretically meaningful.

Using reverse correlation in adults allows us to identify canonical ToM and Social Pain events in a data-driven yet hypothesis-informed manner. We then examine how children’s neural responses to these same events vary with age, neural maturity, and dyadic synchrony. This approach is consistent with prior work in developmental social neuroscience (e.g., Richardson et al., 2018) and offers a valid framework for identifying interpretable social-cognitive events in naturalistic stimuli.

We have now clarified the rationale for using adult-based reverse correlation in the revised manuscript and explicitly stated its advantages for identifying targeted ToM and Social Pain content in the stimulus.

The corresponding changes have been made on pages 17 of the revised manuscript.

“We employed reverse correlation analysis in adults to identify discrete events within the movie that elicited reliable neural responses across participants in ToM and SPM networks.

The events of adults were chosen for this analysis due to the relative stability and maturity of their social brain responses, allowing for robust detection of canonical ToM and social pain-related moments. These events, once identified, served as stimulus-locked timepoints for subsequent analyses in the child cohort. This approach enables us to examine how children's responses to well-characterized, socially meaningful events vary with age and parent-child dyadic dynamics.”

No learning effects or temporal lagged effects are tested in the current study, so the results do not support the authors' conclusions that the data speak to Bandura's social learning theory. The authors do mention theories of biobehavioral synchrony in the introduction but do not discuss this framework in the discussion (which is most directly relevant to the data). The data can also speak to other neurodevelopmental theories of development (e.g.,neuroconstructivist approaches), but the authors do not discuss them. The manuscript would benefit from significantly revising the framework to focus more on biobehavioral synchrony data and other neurodevelopmental approaches given the prior work done in this area rather than a social psychology framework that is not directly evaluated.

We appreciate the reviewer’s thoughtful and constructive feedback. We agree that the current study does not directly test mechanisms central to Bandura’s social learning theory, such as observational learning over time or behavioral modeling. In light of this, we have significantly revised the theoretical framing of the manuscript to focus more directly on the biobehavioral synchrony framework, which more accurately reflects the dyadic neural measures employed in this study and is better supported by our findings.

Specifically, we have expanded the Discussion to contextualize our findings in terms of biobehavioral synchrony, emphasizing how inter-subject neural synchronization may reflect coordinated parent-child engagement and emotional attunement. We have also incorporated insights from neurodevelopmental and neuroconstructivist models, acknowledging that social cognitive development is shaped by dynamic interactions between neural maturation and environmental input over time.

Although we continue to briefly reference Bandura’s theory to situate our findings within broader social-cognitive frameworks, we have clearly delineated the boundaries of what our data can support and have tempered previous claims. These changes are intended to better align our conceptual framing with the empirical evidence and relevant theoretical models.

The corresponding changes have been made on pages 11-12 of the revised manuscript.

“Insights into mechanisms of Neuroconstructivist Perspectives and Bandura’s social learning theory

Our findings align with a neuroconstructivist perspective, which conceptualizes brain development as an emergent outcome of reciprocal interactions between biological constraints and context-specific environmental inputs. Rather than presuming fixed traits or linear maturation, this perspective highlights how neural circuits adaptively organize in response to experience, gradually supporting increasingly complex cognitive functions49. It offers a particularly powerful lens for understanding how early caregiving environments modulate the maturation of social brain networks.

Building on this framework, the present study reveals that moment-to-moment neural synchrony between parent and child, especially during emotionally salient or socially meaningful moments, is associated with enhanced Theory of Mind performance and reduced dyadic conflict. This suggests that beyond age-dependent neural maturation, dyadic neural coupling may serve as a relational signal, embedding real-time interpersonal dynamics into the child’s developing neural architecture [1] . Our data demonstrate that children’s brains are not merely passively maturing, but are also shaped by the relational texture of their lived experiences—particularly interactions characterized by emotional engagement and joint attention. Importantly, this adds a new dimension to neuroconstructivist theory: it is not simply whether the environment shapes development, but how the quality of interpersonal input dynamically calibrates neural specialization. Interpersonal variation leaves detectable signatures in the brain, and our use of neural synchrony as a dyadic metric illustrates one potential pathway through which caregiving relationships exert formative influence on the developing social brain.

The contribution of this work lies not in reiterating the interplay of nature and nurture, but in specifying the mechanistic role of interpersonal neural alignment as a real-time, context-sensitive developmental input. Neural synchrony between parent and child may function as a form of relationally grounded, temporally structured experience that tunes the child’s social brain toward contextually relevant signals. Unlike generalized enrichment, this form of neural alignment is inherently personalized and contingent—features that may be especially potent in shaping social cognitive circuits during early childhood.

Although our study was not designed to directly examine learning mechanisms such as imitation or reinforcement, the findings can be viewed as broadly consistent with social learning theory. Bandura's theory posits that human behavior is shaped by observational learning and modeling from others in one's environment [2-4]. According to Bandura, children acquire social cognitive skills by observing and interacting with their parents and other significant figures in their environment. This dynamic interplay shapes their ability to understand and predict the behavior of others, which is crucial for the development of ToM and other social competencies.”

References

(1) Hughes, C. et al. Origins of individual differences in theory of mind: From nature to nurture? Child development 76, 356-370 (2005).

(2) Koole, S. L. & Tschacher, W. Synchrony in psychotherapy: A review and an integrative framework for the therapeutic alliance. Frontiers in psychology 7, 862 (2016).

(3) Liu, D., Wellman, H. M., Tardif, T. & Sabbagh, M. A. Theory of mind development in Chinese children: a meta-analysis of false-belief understanding across cultures and languages. Developmental Psychology 44, 523 (2008).

(4) Frith, U. & Frith, C. D. Development and neurophysiology of mentalizing. Philosophical Transactions of the Royal Society of London. Series B: Biological Sciences 358, 459-473 (2003).

The significance and impact of the findings would be clearer if the authors more clearly situated the findings in the context of (a) other movie and theory of mind fMRI task data during development; and (b) existing data on parent-child neural synchrony (often uses fNIRS or EEG). What principles of brain and social cognition development do these data speak to? What is new?

We thank the reviewer for this thoughtful comment. In response, we have revised the Discussion section to more clearly situate our findings within two key literatures: (a) fMRI studies examining Theory of Mind using movie-based and traditional task paradigms across development, and (b) research on parent-child neural synchrony. We now articulate more explicitly how our findings advance current understanding of the neural architecture of social cognition in childhood, and how they contribute new insights into the relational processes shaping brain function. These revisions clarify the conceptual and empirical novelty of our study, particularly in its use of naturalistic fMRI, simultaneous child-parent dyads, and integration of neural maturity with interpersonal synchrony.

The corresponding changes have been made on pages 12 of the revised manuscript.

“Our findings contribute to and extend prior research using fMRI paradigms to investigate ToM development in children. Previous work has shown that these networks become increasingly specialized and differentiated throughout childhood [1-3]. The current study extends these findings by demonstrating that the development of social brain networks is a gradual process that continues beyond the preschool years and is related to children's chronological age. This finding is consistent with behavioral research indicating that ToM and social abilities continue to develop and refine throughout middle childhood and adolescence [4]. Importantly, we move beyond prior work by combining reverse correlation with naturalistic stimuli to isolate discrete, behaviorally meaningful events (e.g., mental state attribution, social rejection) and relate children’s brain responses to adult patterns and social outcomes. This event-level analysis in a dyadic context offers greater ecological and interpretive precision than traditional block or condition-based designs. Our study provides novel evidence for the neural underpinnings of this protracted development, suggesting that the functional maturation of social brain networks may support the continued acquisition and refinement of social cognitive skills.

In parallel, our study builds on and extends a growing body of work on parent-child neural synchrony, much of which has relied on fNIRS or EEG hyperscanning to demonstrate interpersonal alignment during communication, shared attention, or cooperative tasks [5-7]. While these modalities offer fine temporal resolution, they are limited in spatial precision and typically focus on surface-level cortical regions such as the prefrontal cortex. By contrast, our naturalistic fMRI approach enables the examination of deep and distributed brain networks—specifically those supporting social cognition—within child-parent dyads during emotionally and cognitively rich scenarios. Intriguingly, we found that neural synchronization during movie viewing was higher in child-mother dyads compared to child-stranger dyads.”

Reference

(1) Jacoby, N., Bruneau, E., Koster-Hale, J. & Saxe, R. Localizing Pain Matrix and Theory of Mind networks with both verbal and non-verbal stimuli. Neuroimage 126, 39-48 (2016).

Astington, J. W. & Jenkins, J. M. A longitudinal study of the relation between language and theory-of-mind development. Developmental Psychology 35, 1311 (1999).

(2) Carter, E. J. & Pelphrey, K. A. School-aged children exhibit domain-specific responses to biological motion. Social Neuroscience 1, 396-411 (2006).

(3) Cantlon, J. F., Pinel, P., Dehaene, S. & Pelphrey, K. A. Cortical representations of symbols, objects, and faces are pruned back during early childhood. Cerebral Cortex 21, 191-199 (2011).

(4) Im-Bolter, N., Agostino, A. & Owens-Jaffray, K. Theory of mind in middle childhood and early adolescence: Different from before? Journal of experimental child psychology 149, 98-115 (2016).

(5) Deng, X. et al. Parental involvement affects parent-adolescents brain-to-brain synchrony when experiencing different emotions together: an EEG-based hyperscanning study. Behavioural brain research 458, 114734 (2024).

(6) Miller, J. G. et al. Inter-brain synchrony in mother-child dyads during cooperation: an fNIRS hyperscanning study. Neuropsychologia 124, 117-124 (2019).

(7) Nguyen, T., Bánki, A., Markova, G. & Hoehl, S. Studying parent-child interaction with hyperscanning. Progress in brain research 254, 1-24 (2020).

There is little discussion about the study limitations, considerations about the generalizability of the findings, and important next steps and future directions. What can the data tell us, and what can it NOT tell us?

We appreciate the reviewer’s recommendation to elaborate on the study’s limitations, generalizability, and future directions. In response, we have added a dedicated section to the Discussion that critically addresses these considerations. We acknowledge the cross-sectional nature of the study, the modest sample size, and the use of a single stimulus context as key limitations. We also clarify the inferences that can be drawn from our data and what remains speculative. Finally, we outline specific future research directions.

The corresponding changes have been made on pages 13-14 of the revised manuscript.

“While leveraging a naturalistic movie-viewing paradigm allowed us to study children's spontaneous neural responses during a semi-structured yet engaging task, dedicated experimental designs are still needed to make stronger inferences about the cognitive processes involved. Additionally, our region-of-interest approach precluded examination of whole-brain networks; future work could explore developmental changes in broader functional circuits. The cross-sectional nature of our study is a further limitation, as it cannot definitively establish the causal directions of the observed relationships. Longitudinal designs tracking children's brain development and social cognitive abilities over time would help clarify whether early parenting impacts later neural maturation and behavioral outcomes, or vice versa. Our sample was restricted to mother-child dyads, leaving open questions about potential differences in father-child relationships and gender effects on parenting neurobiology. Larger and more diverse samples would enhance the generalizability of the findings.

Several future directions emerge from this research. First, combining naturalistic neuroimaging with structured cognitive tasks could elucidate the specific mental processes underlying children's neural responses during movie viewing. Examining how these processes relate to real-world social behavior would further bridge neurocognitive function and ecological validity. Longitudinal studies beginning in infancy could chart the developmental trajectories of parent-child neural synchrony and their impact on long-term social outcomes. Such work could also explore sensitive periods when parenting may be most influential on social brain maturation. Finally, expanding this multimodal approach to clinical populations like autism could yield insights into atypical social cognitive development and inform tailored intervention strategies targeting parent-child relationships and neural plasticity.”

To evaluate associations between child neural activity patterns during the movie AND parent-child synchronization patterns AND other variables such as parent-child communication and theory of mind behavior, it seems like a robust approach could be to examine whether similar synchronization patterns are associated with similar scores on different variables. Would allow for non-linear and multivariate associations.

We greatly appreciate the reviewer’s thoughtful suggestion regarding the use of similarity-based or multivariate analyses to assess whether dyads with similar neural synchronization profiles also exhibit similar scores on behavioral or relational variables. We agree that this type of analysis—such as representational similarity analysis (RSA) or inter-subject pattern similarity—offers a powerful framework for capturing non-linear and multivariate associations, and could provide deeper insights into shared neurobehavioral patterns across participants. However, the analytic logic of similarity-based approaches typically requires the availability of comparable measures across individuals or dyads (e.g., child A and child B must both have measures of brain activity, behavior, and environment). In the present study, our focus was on the child as the behavioral and developmental target, and we did not collect parallel behavioral or cognitive variables from the parent side (e.g., adult Theory of Mind ability, emotional traits, parenting style questionnaires beyond dyadic reports). As a result, it was not feasible to construct pairwise similarity matrices across dyads that include both neural synchrony and matched behavioral dimensions from both individuals.

Instead, our study was designed to examine how child-level outcomes (e.g., Theory of Mind performance, social functioning) are associated with (a) the child’s neural responses to specific social events, and (b) the degree of neural synchronization with their mother, as a marker of relational engagement. The analytical emphasis, therefore, remained on within-child variation, modulated by the quality of the parent-child interaction.

Were there associations between parent-child neural synchronization and child age? What was the association between neural maturity and parent-child neural synchronization

We thank the reviewer for raising this important point regarding associations between parent-child neural synchronization (ISS), child age, and neural maturity.

As reported in the original manuscript, we did not observe significant correlations between parent-child ISS and child age for either the Theory of Mind (ToM) or Social Pain Matrix (SPM) networks (all ps > 0.1). Additionally, we conducted additional analysis, we found no significant correlations between ISS and neural maturity (Author response image 1, r = 0.2503, p = 0.1533).

These findings indicate that parent-child neural synchronization in this naturalistic viewing context is not simply explained by age-related maturation or children's neural maturity level. Instead, ISS may predominantly reflect real-time interpersonal engagement or relational dynamics rather than individual developmental trajectories or neural maturity.

**Author response image 1. sa4fig1:** Scatterplot showing the association between parent-child inter-subject synchronization (ISS) and neural maturity, averaged across the Theory of Mind (ToM) and Social Pain Matrix (SPM) networks. Each point represents one dyad. No significant correlation was observed between ISS and neural maturity (r = 0.2503, p = 0.1533), suggesting that interpersonal neural synchronization and individual neural maturation may reflect dissociable aspects of social brain development.

The rationale for splitting the ages into 3 groups is unclear and creates small groups that could be more prone to spurious associations. Why not look at age continuously?

We thank the reviewer for raising this important point. We fully agree that analyzing age as a continuous variable is statistically more robust and minimizes concerns about spurious associations due to arbitrary groupings.

To clarify, all primary statistical models—including correlational analyses—treated age as a continuous variable, and our core developmental inferences are based on these continuous-age findings.

In addition to these analyses, we included age group comparisons as a supplementary approach, guided by both theoretical considerations and visual inspection of the data. Specifically, we aimed to explore whether functional differentiation between social brain networks (e.g., ToM and SPM) might begin to emerge non-linearly or earlier than expected, particularly in the youngest children. Such early neural divergence may not be well-captured by linear trends alone. The grouped analysis allowed us to illustrate that network differentiation was already observable in children under age 5, suggesting that certain aspects of social brain organization may emerge earlier than classically assumed.

We have now clarified this rationale in the revised manuscript and emphasized that the group-based analysis was used solely to highlight developmental shifts that may not follow a linear pattern, and not for formal hypothesis testing.

The corresponding changes have been made on pages 9 of the revised manuscript.

“While our primary analyses treated age as a continuous variable, we also performed exploratory group-based comparisons to probe for potential non-linear developmental shifts in social brain network organization. This approach revealed that the differentiation between ToM and SPM networks was already present in the youngest group (ages 3–4), suggesting that early neural specialization may begin prior to the age at which ToM behavior is reliably observed. These group-level observations provide complementary evidence to the continuous analyses and may inform future work examining sensitive periods or early markers of social brain development.”

Tables would be improved if they were more professionally formatted (e.g., names of the variables rather than variable abbreviation codes).

We appreciate the reviewer’s suggestion to improve the clarity and professionalism of our tables. In the revised manuscript, we have reformatted all tables to include full variable names rather than abbreviations or coded labels, and we ensured consistency in terminology across the manuscript text, tables, and figure legends. We have also added explanatory footnotes where needed to clarify any derived or composite measures. We hope these revisions improve the accessibility and readability of the results for a broader audience

**Reviewer #2:**
Summary:This study investigates the impact of mother-child neural synchronization and the quality of parent-child relationships on the development of Theory of Mind (ToM) and social cognition. Utilizing a naturalistic fMRI movie-viewing paradigm, the authors analyzed inter-subject neural synchronization in mother-child dyads and explored the connections between neural maturity, parental caregiving, and social cognitive outcomes. The findings indicate age-related maturation in ToM and social pain networks, emphasizing the importance of dyadic interactions in shaping ToM performance and social skills, thereby enhancing our understanding of the environmental and intrinsic influences on social cognition.Strengths:This research addresses a significant question in developmental neuroscience, by linking social brain development with children's behaviors and parenting. It also uses a robust methodology by incorporating neural synchrony measures, naturalistic stimuli, and a substantial sample of mother-child dyads to enhance its ecological validity. Furthermore, the SEM approach provides a nuanced understanding of the developmental pathways associated with Theory of Mind (ToM).

We appreciate the positive evaluation and valuable comments of the reviewer. According to the reviewer`s comments, we have revised the manuscript thoroughly to address the concerns raised by the reviewer. A point-by-point response to each of the issues raised by the reviewer has been made. We believe that the revision of our manuscript has now been significantly improved.

Upon reviewing the introduction, I feel that the first goal - developmental changes of the social brain and its relation to age - seems somewhat distinct from the other two goals and the main research question of the manuscript. The authors might consider revising this section to enhance the overall coherence of the manuscript. Additionally, the introduction lacks a clear background and rationale for the importance of examining age-related changes in the social brain.

We thank the reviewer for this thoughtful observation. In response, we have revised the Introduction to better integrate the developmental aspect of the social brain with the broader research aims. We now explicitly link age-related changes in social brain organization to the emergence of social cognitive abilities and highlight why early childhood (ages 3–8) represents a particularly formative period. This revision clarifies that our first aim—examining functional specialization and neural maturity in Theory of Mind (ToM) and Social Pain Matrix (SPM) networks—serves as a developmental foundation for understanding how dyadic influences, such as neural synchrony and caregiving quality, shape children’s social cognition.

We have also improved the rationale for examining age-related change, drawing on key literature in developmental neuroscience to show how the early emergence and specialization of social brain networks provide a necessary context for interpreting interpersonal neural dynamics.

The corresponding changes have been made on pages 3 of the revised manuscript.

“These findings suggest that the development of specialized brain regions for reasoning about others' mental states and physical sensations is a gradual process that continues throughout childhood.

Understanding how these networks differentiate with age is essential not only for mapping typical brain development, but also for contextualizing the role of environmental influences. By establishing normative patterns of neural maturity and differentiation, we can better interpret how relational experiences—such as caregiver-child synchrony and parenting quality—modulate these trajectories. Thus, our first goal provides a developmental anchor that grounds our investigation of interpersonal and environmental contributions to social brain function.”

The manuscript uses both "mother-child" and "parent-child" terminology. Does this imply that only mothers participated in the fMRI scans while fathers completed the questionnaires? If so, have the authors considered the potential impact of parental roles (father vs. mother)?

We thank the reviewer for raising this important point regarding terminology and parental roles. To clarify, all participating caregivers in the current study were biological mothers, and all behavioral questionnaires were also completed by these same mothers. No fathers were included in this study. We have revised the manuscript throughout to consistently use the term “mother-child” when referring to the specific dyads in our sample.

We also appreciate the opportunity to elaborate on the rationale for including only mothers. Prior research has shown that maternal and paternal influences on child development are not interchangeable, and that the neural correlates of caregiving behaviors differ between mothers and fathers. For example, studies have demonstrated distinct patterns of brain activation during social and emotional processing in mothers versus fathers (Abraham et al., 2014; JE Swain et al., 2014). Given these differences, we deliberately focused on mother-child dyads to maintain neurobiological consistency in our analysis and reduce variance associated with heterogeneous caregiving roles. We now clarify this rationale in the revised Methods and Discussion sections.

The corresponding changes have been made on pages 14 of the revised manuscript.

“We chose to focus exclusively on mother-child dyads in this study based on prior evidence suggesting distinct neural and behavioral caregiving profiles between mothers and fathers [1-2], allowing us to maintain role consistency and reduce variability in dyadic interactions.

Our sample was restricted to mother-child dyads, leaving open questions about potential differences in father-child relationships and gender effects on parenting neurobiology [1]. Larger and more diverse samples would enhance the generalizability of the findings.”

Reference:

(1) Swain, J. E. et al. Approaching the biology of human parental attachment: Brain imaging, oxytocin and coordinated assessments of mothers and fathers. Brain research 1580, 78-101 (2014).

(2) Abraham, E. et al. Father's brain is sensitive to childcare experiences. Proceedings of the National Academy of Sciences 111, 9792-9797 (2014).

There is inconsistent usage of the terms ISC and ISS in the text and figures, both of which appear to refer to synchronization derived from correlation analysis. It would be beneficial to maintain consistency throughout the manuscript.

We thank the reviewer for highlighting the inconsistent use of “ISC” and “ISS” in the original manuscript. We agree that clarity and consistency in terminology are essential. In response, we have revised the manuscript to consistently use “ISS” (inter-subject synchronization) throughout the text, figures, tables, and legends.

Of the 50 dyads, 16 were excluded due to data quality issues, which constitutes a significant proportion. It would be helpful to know whether these excluded dyads exhibited any distinctive characteristics. Providing information on demographic or behavioral differences-such as Theory of Mind (ToM) performance and age range between the excluded and included dyads would enhance the assessment of the findings' generalizability.

We thank the reviewer for this important observation. We agree that understanding the characteristics of excluded participants is essential for assessing the generalizability of the findings.

In response, we conducted comparative analyses between included and excluded dyads (N = 34 included; N = 16 excluded) on key demographic and behavioral variables, including child age, gender, and Theory of Mind (ToM) performance. These analyses revealed no significant differences between groups on any of these measures (ps > 0.1), suggesting that data exclusion due to quality issues (e.g., excessive motion, incomplete scans) did not introduce systematic bias.

We have now added this information to the Results and Methods sections of the manuscript.

The corresponding changes have been made on pages 6 and 17 of the revised manuscript.

“Of the 50 initial mother-child dyads recruited, 16 were excluded due to excessive head motion (n = 11), incomplete scan sessions (n = 3), or technical issues during data acquisition (n = 2). The final sample consisted of 34 dyads. To assess potential bias introduced by data exclusion, we compared included and excluded dyads on child age, gender, and Theory of Mind performance. No significant differences were found across these variables (all ps > 0.1), suggesting that the analytic sample was demographically representative of the full cohort.

Comparison between included and excluded dyads revealed no significant differences in child age (t = 1.23, p = 0.24), ToM scores (t = -0.54, p = 0.59), or sex distribution (χ² < 0.01, p = 0.98), indicating that data exclusion did not bias the sample in a systematic way.”

The article does not adhere to the standard practice of using a resting state as a baseline for subtracting from task synchronization. Is there a rationale for this approach? Not controlling for a baseline may lead to issues, such as whether resting state synchronization already differs between subjects with varying characteristics.

We thank the reviewer for raising this important methodological point. We agree that controlling for baseline synchronization, such as using a resting-state scan as a comparison, can help disambiguate whether task-induced synchrony reflects genuine stimulus-driven coupling or baseline differences across individuals or dyads.

In the present study, we focused on inter-subject synchronization (ISS) during naturalistic movie viewing, a task condition that has been widely used in previous developmental and social neuroscience research to assess shared neural engagement. We did not include a resting-state scan in the current protocol due to time constraints and the young age of our participants (ages 3–8), as longer scanning sessions often result in increased motion and reduced data quality in pediatric populations. Moreover, many prior studies using ISS in naturalistic paradigms have similarly focused on task-driven synchrony without subtracting a resting baseline (e.g., Hasson et al., 2004; Nguyen et al., 2020; Reindl et al., 2018).

That said, we acknowledge that baseline neural synchrony across dyads may vary depending on individual or relational characteristics (e.g., temperament, arousal, attentional style), and this remains an important question for future research. In the revised Discussion, we now explicitly note the absence of a resting-state baseline as a limitation and highlight the need for future studies to examine how resting and task-based ISS may interact, particularly in the context of child-caregiver dyads.

The corresponding changes have been made on page 13 of the revised manuscript.

“Another limitation of the current design is the lack of a resting-state baseline for inter-subject synchronization. While our focus was on synchronization during naturalistic social processing, we cannot determine whether individual differences in ISS reflect purely task-induced coupling or are partially shaped by trait-level synchrony present at rest. Including both resting and task conditions in future work would allow for stronger inferences about stimulus-specific versus baseline-driven synchronization, especially in relation to interpersonal factors such as relationship quality or social responsiveness.”

The title of the manuscript suggests a direct influence of mother-child interactions on children's social brain and theory of mind. However, the use of structural equation modeling (SEM) may not fully establish causal relationships. It is possible that the development of children's social brain and ToM also enhances mother-child neural synchronization. The authors should address this alternative hypothesis of the potential bidirectional relationship in the discussion and exercise caution regarding terms that imply causality in the title and throughout the manuscript.

We appreciate the reviewer’s careful attention to issues of causality in our manuscript. We agree that our cross-sectional design limits causal inference, and that the use of structural equation modeling (SEM) in this context does not allow for conclusions about directional or mechanistic pathways. In response, we have revised the Discussion to explicitly acknowledge these limitations, and now include an expanded section on the potential for bidirectional or co-constructed processes, consistent with neuroconstructivist frameworks.

We have also tempered the interpretation of our SEM findings, avoiding causal language throughout the manuscript and clarifying that our analyses are exploratory and associational in nature. We hope that these changes provide a more cautious and developmentally grounded interpretation of the data.

With regard to the title, we respectfully chose to retain the original wording, as we believe it captures the thematic focus and central research question of the paper—namely, the potential role of mother-child interaction in the development of children’s social brain and Theory of Mind. While we understand the reviewer’s concern, we note that the interpretation of this phrasing is contextualized within the manuscript, which now includes clear qualifications regarding the limits of causal inference. We have taken care to ensure that no claims of unidirectional causality are made in the body of the paper.

The corresponding changes have been made on pages 11- 12 of the revised manuscript.

“Our findings align with a neuroconstructivist perspective, which conceptualizes brain development as an emergent outcome of reciprocal interactions between biological constraints and context-specific environmental inputs. Rather than presuming fixed traits or linear maturation, this perspective highlights how neural circuits adaptively organize in response to experience, gradually supporting increasingly complex cognitive functions54. It offers a particularly powerful lens for understanding how early caregiving environments modulate the maturation of social brain networks.

Building on this framework, the present study reveals that moment-to-moment neural synchrony between parent and child, especially during emotionally salient or socially meaningful moments, is associated with enhanced Theory of Mind performance and reduced dyadic conflict. This suggests that beyond age-dependent neural maturation, dyadic neural coupling may serve as a relational signal, embedding real-time interpersonal dynamics into the child’s developing neural architecture. Our data demonstrate that children’s brains are not merely passively maturing, but are also shaped by the relational texture of their lived experiences—particularly interactions characterized by emotional engagement and joint attention. Importantly, this adds a new dimension to neuroconstructivist theory: it is not simply whether the environment shapes development, but how the quality of interpersonal input dynamically calibrates neural specialization. Interpersonal variation leaves detectable signatures in the brain, and our use of neural synchrony as a dyadic metric illustrates one potential pathway through which caregiving relationships exert formative influence on the developing social brain.

The contribution of this work lies not in reiterating the interplay of nature and nurture, but in specifying the mechanistic role of interpersonal neural alignment as a real-time, context-sensitive developmental input. Neural synchrony between parent and child may function as a form of relationally grounded, temporally structured experience that tunes the child’s social brain toward contextually relevant signals. Unlike generalized enrichment, this form of neural alignment is inherently personalized and contingent—features that may be especially potent in shaping social cognitive circuits during early childhood.

The cross-sectional nature of our study is a further limitation, as it cannot definitively establish the causal directions of the observed relationships. Longitudinal designs tracking children's brain development and social cognitive abilities over time would help clarify whether early parenting impacts later neural maturation and behavioral outcomes, or vice versa.”

I would appreciate more details about the 14 Theory of Mind (ToM) tasks, which could be included in supplemental materials. The authors score them on a scale from 0 to 14 (each task 1 point); however, the tasks likely vary in difficulty and should carry different weights in the total score (for example, the test and the control questions should have different weights). Many studies have utilized the seven tasks according to Wellman and Liu (2004), categorizing them into "basic ToM" and "advanced ToM." Different components of ToM could influence the findings of the current study, which should be further examined by a more in-depth analysis.

We thank the reviewer for raising this important point regarding the structure and scoring of the Theory of Mind (ToM) tasks. We will provide a detailed description of all 14 tasks in the Supplemental Materials, including their content, targeted mental state concepts (e.g., beliefs, desires, intentions), and design features (e.g., test/control items, task format).

We fully agree that ToM tasks differ in complexity, and in principle, a weighted or component-based scoring approach (e.g., distinguishing basic and advanced ToM) could offer greater interpretive value. However, in our study design, tasks were administered in a fixed sequence from lower to higher difficulty, and testing was terminated if the child was unable to successfully complete three consecutive tasks. This approach is developmentally appropriate for younger children but results in non-random missingness for more advanced tasks—particularly among children at the lower end of the age range (3–4 years).

Given this adaptive task structure, re-scoring using weighted or subscale-based approaches would introduce systematic bias, as children who struggled with early items were not administered more complex ones. As a result, a full breakdown by task type (e.g., basic vs. advanced ToM) would only reflect a restricted subsample and would not be comparable across the full cohort. For this reason, we retained the unit-weighted total ToM score as the most developmentally valid and comparable metric across participants.

**Reviewer #3:**
Summary:The article explores the role of mother-child interactions in the development of children's social cognition, focusing on Theory of Mind (ToM) and Social Pain Matrix (SPM) networks. Using a naturalistic fMRI paradigm involving movie viewing, the study examines relationships among children's neural development, mother-child neural synchronization, and interaction quality. The authors identified a developmental pattern in these networks, showing that they become more functionally distinct with age. Additionally, they found stronger neural synchronization between child-mother pairs compared to child-stranger pairs, with this synchronization and neural maturation of the networks associated with the mother-child relationship and parenting quality.Strengths:This is a well-written paper, and using dyadic fMRI and naturalistic stimuli enhances its ecological validity, providing valuable insights into the dynamic interplay between brain development and social interactions. However, I have some concerns regarding the analysis and interpretation of the findings. I have outlined these concerns below in the order they appear in the manuscript, which I hope will be helpful for the revision.

We appreciate the reviewer’s thoughtful and constructive summary of the manuscript. The concerns raised regarding aspects of the analysis and interpretation have been carefully considered. Detailed point-by-point responses are provided below, along with descriptions of the corresponding revisions made to improve the clarity, precision, and interpretive caution of the manuscript.

Given the importance of social cognition in this study, please cite a foundational empirical or review paper on social cognition to support its definition. The current first citation is primarily related to ASD research, which may not fully capture the broader context of social cognition development.

We thank the reviewer for this helpful suggestion. We agree that a broader, foundational reference is more appropriate for introducing the concept of social cognition. In response, we have revised the Introduction to include a widely cited theoretical or review paper on social cognition to provide a more general developmental context.

The corresponding changes have been made on pages 3 of the revised manuscript.

“Social cognition, defined as the ability to interpret and predict others' behavior based on their beliefs and intentions and to interact in complex social environments and relationships is a crucial aspect of human development [1-2]”

(1) Adolphs, R. The social brain: neural basis of social knowledge. Annual review of psychology 60, 693-716 (2009).

(2) Frith, C. D. & Frith, U. Mechanisms of social cognition. Annual review of psychology 63, 287-313 (2012).

It is standard practice to report the final sample size in the Abstract and Introduction, rather than the initial recruited sample, as high attrition rates are common in pediatric studies. For example, this study recruited 50 mother-child dyads, and only 34 remained after quality control. This information is crucial for interpreting the results and conclusions. I recommend reporting the final sample size in the abstract and introduction but specifying in the Methods that an additional 16 mother-child dyads were initially recruited or that 50 dyads were originally collected.

We thank the reviewer for this helpful recommendation. In the original version of the manuscript, the Abstract and Introduction referenced the total number of dyads recruited (N = 50). In line with standard reporting practices and to ensure clarity regarding the analytic sample, we have now revised both the Abstract and Introduction to report the final sample size (N = 34). The full recruitment and exclusion details—including the number of dyads removed due to excessive motion or technical issues—are now clearly described in the Methods section.

The corresponding changes have been made on pages 1 and 4 of the revised manuscript.

In the "Neural maturity reflects the development of the social brain" section, the authors report the across-network correlation for adults, finding a negative correlation between ToM and SPM. However, the cross-network correlations for the three child groups are not reported. The statement that "the two networks were already functionally distinct in the youngest group of children we tested" is based solely on within-network positive correlations, which does not fully demonstrate functional distinctness. Including cross-network correlations for the child groups would strengthen this conclusion.

We thank the reviewer for this insightful comment. We agree that within-network correlations alone do not fully establish functional distinctness, particularly in early development. To more directly test whether the ToM and SPM networks were already differentiated in children, we have now included the cross-network correlations between the two networks for each of the three age groups in the revised manuscript. These findings support and strengthen our original claim that the ToM and SPM networks are functionally dissociable even in early childhood, and we have revised the relevant Results sections accordingly to reflect this.

The corresponding changes have been made on page 7 of the revised manuscript.

“In children, each network also exhibited positive correlations within-network and negative correlations across networks within-ToM correlation M(s.e.) = 0.31(0.04); within-SPM correlation M(s.e.) = 0.29(0.04); across-network M(s.e.) = −0.09 (0.02).

In the Pre-junior group only (3-4 years old children, n = 12), both ToM and SPM networks had positive within-network correlations (within-ToM correlation M (s.e.) = 0.29(0.06); within-SPM correlation M(s.e.) = 0.23(0.05), across-network M(s.e.) = −0.05(0.02)).”

The ROIs for the ToM and SPM networks are defined based on previous literature, applying the same ROIs across all age groups. While I understand this is a common approach, it's important to note that this assumption may not fully hold, as network architecture can evolve with age. The functional ROIs or components of a network might shift, with regions potentially joining or exiting a network or changing in size as children develop. For instance, Mark H. Johnson's interactive specialization theory suggests that network composition may adapt over developmental stages. Although the authors follow the approach of Richardson et al. (2018), it would be beneficial to discuss this limitation in the Discussion. An alternative approach would be to apply data-driven analysis to justify the selection of the ROIs for the two networks.

We thank the reviewer for this thoughtful and theoretically grounded comment. In our study, we followed the approach of Richardson et al. (2018), using a priori ROIs defined from adult meta-analyses and ToM/SPM task studies. This approach facilitates comparison with prior work and provides anatomical consistency across participants. However, we fully agree that applying adult-defined ROIs to pediatric populations involves important assumptions about the stability of network architecture across development, which may not fully hold in early childhood.

We have now addressed this limitation more explicitly in the revised Discussion, emphasizing that the fixed-ROI approach may not capture the dynamic reorganization of social brain networks during development.

The corresponding changes have been made on pages 13 of the revised manuscript.

“Moreover, the ROIs used to define the ToM and SPM networks were based on meta-analyses and task studies primarily conducted with adults. While this approach promotes comparability with existing literature, it assumes that the spatial organization of these networks is stable across age groups. However, theories of interactive specialization suggest that the composition and boundaries of functional networks may undergo reorganization during development, with regions potentially entering or exiting networks based on experience and maturational processes. As a result, the current analysis may not fully capture age-specific functional architecture, particularly in younger children. Future studies using data-driven or age-appropriate parcellation methods could provide more precise characterizations of how social brain networks are constructed and differentiated throughout childhood.”

The current sample size (N = 34 dyads) is a limitation, particularly given the use of SEM, which generally requires larger samples for stable results. Although the model fit appears adequate, this does not guarantee reliability with the current sample size. I suggest discussing this limitation in more detail in the Discussion.

We thank the reviewer for highlighting the limitations of applying structural equation modeling (SEM) with a relatively modest sample size. We agree that SEM generally benefits from larger samples to ensure model stability and parameter reliability, and that satisfactory model fit does not guarantee robustness in small-sample contexts.

In the revised Discussion, we now more clearly acknowledge that the use of SEM in the current study is exploratory in nature, and that all results should be interpreted with caution due to potential sample size-related constraints. The model was constructed to provide an integrated view of the observed associations rather than to establish definitive pathways. We have also added a note that future research with larger samples and longitudinal designs will be needed to validate and extend the proposed model.

The corresponding changes have been made on pages 13 of the revised manuscript.

“In addition, the modest sample size (N = 34 dyads) presents limitations for the application of structural equation modeling (SEM), which typically requires larger samples for stable estimation and generalizable inferences. While the model fit was acceptable, the results should be interpreted as exploratory and hypothesis-generating, rather than confirmatory. Future studies with larger, independent samples will be important for validating the structure and directionality of the proposed relationships”

Based on the above comment, I believe that conclusions regarding the relationship between social network development, parenting, and support for Bandura's theory should be tempered. The current conclusions may be too strong given the study's limitations.

We thank the reviewer for this important and balanced observation. We agree that the conclusions drawn from the current study should reflect the exploratory nature of the analyses, as well as the methodological limitations, including the modest sample size and cross-sectional design.

In response, we have revised the Conclusion sections to use more cautious, associative language when describing the observed relationships among social brain development, parenting factors, and Theory of Mind outcomes. In particular, we have tempered statements regarding support for Bandura’s social learning theory, clarifying that while our findings are consistent with social learning frameworks, the data do not allow for direct tests of modeling or observational learning mechanisms.

We hope these revisions help clarify the scope of the findings and improve the conceptual rigor of the manuscript.

The corresponding changes have been made on pages 14 of the revised manuscript.

“Our study provides novel evidence that children's social cognitive development may be shaped by the intricate interplay between environmental influences, such as parenting, and biological factors, such as neural maturation. Our findings contribute to a growing understanding of the factors associated with social cognitive development and suggest the potential importance of parenting in this process. Specifically, the study points to the possible role of the parent-child relationship in supporting the development of social brain circuitry and highlights the relevance of family-based approaches for addressing social difficulties. The observed neural synchronization between parent and child, which was associated with relationship quality, underscores the potential significance of positive parental engagement in fostering social cognitive skills. Future longitudinal and clinical research can build on this multimodal approach to further clarify the neurobehavioral mechanisms underlying social cognitive development. Such research may help inform more effective strategies for promoting healthy social functioning and mitigating social deficits through targeted family-based interventions.”

The SPM (pain) network is associated with empathic abilities, also an important aspect of social skills. It would be relevant to explore whether (or explain why) SPM development and child-mother synchronization are (or are not) related to parenting and the parent-child relationship.

We thank the reviewer for this thoughtful and important comment regarding the role of the Social Pain Matrix (SPM) network in social cognition and empathy. We agree that this network represents a critical component of social-cognitive development and is theoretically linked to affective processing and interpersonal understanding.

We would like to clarify that in our existing analyses—already included in the original submission and detailed in the Supplemental Results—SPM network measures showed similar significant associations with behavioral outcomes than the ToM network. These outcomes included children's performance on ToM tasks as well as broader measures of social functioning. We have added more discussion in the supplementary results.

“To further investigate the specificity of our findings, we conducted additional control analyses focusing on the individual components of the social brain networks examined in our study: the Theory of Mind (ToM) and Social Pain Matrix (SPM) networks.

When analyzing these networks separately, we found significant correlations between neural maturity and age, as well as between inter-subject synchronization (ISS) and parent-child relationship quality for both the ToM and SPM networks individually (Fig. S1). Specifically, neural maturity within each network was positively correlated with age, indicating that both networks undergo maturation during childhood. Similarly, ISS within each network was negatively correlated with parent-child conflict scores, suggesting that both networks contribute to the observed relationship between neural synchrony and parent-child relationship quality.

These results highlight the importance of considering the social brain as an integrated system, where the ToM and SPM networks work in concert to support social cognitive development. While each network shows age-related maturation and sensitivity to parent-child relationship quality, their combined functioning appears to be crucial for predicting broader social cognitive outcomes.